# Cobamide-producing microbes as a model for understanding general nutritional interdependencies in soil food webs

Qi Zhang [1,2], Bingfeng Chen[3], Zhenyan Zhang[1,2], Yitian Yu[3], Mingkang Jin[4], Tao Lu [3], Ziyao Zhang[3], Qian Pang[3], Nuohan Xu [1,2], Jianqiang Sun[3], Jun Chen [5], Jichen Wang[4], Dong Zhu [6] ✉, Haifeng Qian [1,3] ✉, Josep Penuelas [7,8] & Yong-Guan Zhu [4,6]

Nutrient crossfeeding critically governs microbiome–host interactions and ecosystem stability. Cobamides, synthesized only by prokaryotes, offer a powerful and tractable model for studying nutrient-mediated interdependencies in soil food webs; however, their ecological role in sustaining soil health remains unclear. Here, we construct the Soil Cobamide Producer database (SCP v.1.0) by integrating over 48,000 metagenomic and genomic datasets from 1,123 sampling sites. This database catalogs phylogenetically diverse prokaryotes (19 phyla, 302 genera) with cobamide biosynthetic potential. Using this resource, we identify host-specific colonization patterns of cobamide-producing microbes in fauna. These microbes also carry diverse functional traits that may contribute to trophic cascades and microbial community stability. In an Enchytraeid model, these colonizers support host development, modulate gene expression, and promote gut stability through transkingdom interactions, with cobamide biosynthesis serving as one representative trait among multiple microbial functions. At macroecological scales, cobamide-producing microbes occur across relatively high trophic levels, reflecting a broader principle of nutrient transfer that may also apply to other essential metabolites. This framework provides a general basis for studying nutritional microbes in soil food webs and advances One Health research.

Nutritional interdependence underlies the stability of both host–microbiome systems and the ecosystems they inhabit. For example, mycorrhizal fungi increase plant phosphorus uptake[1], nitrogen-fixing bacteria supply bioavailable nitrogen to plants[2], and lactic acid bacteria contribute vitamins that influence host–microbe interactions[3]. Among such nutritional interdependencies, cobamides, a family of enzyme cofactors best exemplified by cobalamin (Vitamin $B_{12}$), are required for many essential metabolic pathways[4], yet their biosynthesis is restricted to a limited subset of prokaryotes[5,6]. These prokaryotes engage in nutritional exchanges with a broad range of cobamide-dependent organisms, including most microorganisms and all higher-order animals[7,8]. Cobamide-producing microbes therefore

[1]Institute for Advanced Study, Shaoxing University, Shaoxing, PR China. [2]College of Chemistry & Chemical Engineering, Shaoxing University, Shaoxing, PR China. [3]College of Environment, Zhejiang University of Technology, Hangzhou, PR China. [4]State Key Laboratory of Urban and Regional Ecology, Research Center for Eco-environmental Sciences, Chinese Academy of Sciences, Beijing, PR China. [5]Laboratory of Pollution Exposure and Health Intervention Technology, Interdisciplinary Research Academy, Zhejiang Shuren University, Hangzhou, PR China. [6]Key Laboratory of Urban Environment and Health, Institute of Urban Environment, Chinese Academy of Sciences, Xiamen, PR China. [7]CSIC, Global Ecology Unit CREAF-CSIC-UAB, Barcelona, Catalonia, Spain. [8]CREAF, Campus Universitat Autònoma de Barcelona, Barcelona, Catalonia, Spain. ✉e-mail: dzhu@iue.ac.cn; hfqian@usx.edu.cn

constitute a major source of essential micronutrients for their hosts and associated communities[9–11]. This role, while not unique to cobamides, makes them a tractable model for investigating nutrient interdependencies within food webs. Nevertheless, the ecological and physiological impacts of cobamide-producing microbes, particularly across trophic levels and in shaping soil food web dynamics, remain poorly understood.

Soils harbor the most diverse and complex microbiomes on Earth, serving as major reservoirs of genetic resources with broad implications for human health and biotechnology[12]. Early research demonstrated that Vitamin B$_{12}$ could substitute for soil extracts to stimulate bacterial growth[13], underscoring its ecological importance within microbial communities. More recently, global metagenomic surveys of 2979 soil samples estimated that 56.4% of metagenome-assembled genomes (MAGs) are potential cobamide-producing microbes[14], with nearly 70% of the soil bacterial genera across over 40 soil types relying on cobamides[15]. Together, these findings establish soil as a key reservoir of cobamide-producing microorganisms. In addition to microbes, genomic analyses revealed that approximately 98% of soil fauna encode at least one cobamide-dependent enzyme (Supplementary Fig. 1), suggesting that cobamide-producing microbes contribute not only to microbial interactions but also to the nutritional requirements of animals[16]. Despite growing insights into their diversity and ecology[14,16], the distribution of cobamide-producing microbes within soil food webs and their roles in mediating cross-trophic interactions remain insufficiently resolved.

The soil food web is a cornerstone of the One Health framework, mediating the exchange and cycling of pathogens and beneficial microbes among the soil, humans, and animals[12]. This bidirectional transmission results in the formation of a complex microbial network that not only spreads pathogens but also delivers health-promoting microbes. Soil microorganisms can reach humans and animals through dust inhalation, environmental exposure, and geophagia, establishing symbiotic relationships with both beneficial and detrimental health outcomes[17–19]. While recent One Health research has largely emphasized zoonotic pathogens to mitigate soil-borne risks, the beneficial roles of soil microbiomes, including their capacity to supply essential nutrients, have received less attention[20–22]. Among these, cobamide-producing microbes exemplify how a shared metabolic trait can serve as a representative case study for understanding the principles of cross-kingdom nutrient exchange in soil ecosystems.

In this study, we combine large-scale data analysis, field surveys, and experimental validation to investigate cobamide-producing microbes as a model for nutrient-mediated interactions in soil food webs (Fig. 1). We construct a comprehensive Soil Cobamide Producer (SCP v.1.0) database from 53,276 soil genomes, characterizing their phylogenetic diversity, genomic features, and habitat distributions. To assess their ecological roles within soil food webs, we integrate extensive field sampling with soil microcosm experiments to track functional dynamics across trophic levels and further conduct artificial colonization assays to evaluate their nutritional contributions to hosts. Finally, we place these findings from a broader food web perspective, linking cobamide-producing microbes to higher trophic levels and cross-ecosystem dissemination. This study establishes a molecular resource for cobamide-producing microbes and underscores their potential as a model system for elucidating nutrient-mediated connections linking environmental and animal health.

## Results and discussion
### Cobamide biosynthesis potential in diverse and habitat-specific soil microbes
Genome-resolved metagenomics has expanded access to previously unknown bacterial[23], archaeal[24], and viral[25] genomes, revealing the vast "microbial dark matter" in soil—microorganisms that remain largely uncharacterized and uncultivable[26]. To extend soil genomic resources,

we recovered 7874 medium-quality MAGs from 2727 public metagenomes across 567 sites worldwide, spanning 24 countries and six continents (Supplementary Data 1, 2 and Supplementary Fig. 2a–c). These MAGs presented high novelty, with only 0.5%, 1.2%, and 7% overlap with the Genome Taxonomy Database (85,205 genomes), the NCBI soil genome collection (5363 genomes), and the SMAG database (40,039 genomes), respectively (Supplementary Fig. 2d), highlighting the need for expanded soil metagenomic datasets. By integrating these MAGs with public resources, we compiled a consolidated soil genome compendium of 53,276 genomes (Supplementary Fig. 3), providing a foundation for the systematic exploration of cobamide-producing microbes and other functional groups in soils.

To build on this resource, we applied a stringent MAG-based host assignment approach to minimize false positives in the prediction of cobamide biosynthesis potential[22], particularly those arising from contig incompleteness or contamination[5,14,15]. This analysis identified 4340 MAGs with cobamide biosynthesis capacity, classified as very likely (29%), likely (15%), or possible (55%) producers (Fig. 2a and Supplementary Data 3). These MAGs represented 19 phyla and 302 genera (Fig. 2b), with major contributions from *Proteobacteria* (44.3%), *Actinobacteria* (28.6%), *Firmicutes* (7.2%), and the archaeal group *Thermoproteota* (7.0%). They were distributed across 1123 terrestrial sites, encompassing all climate zones and extreme environments such as permafrost, tundra, glaciers, and deserts (Supplementary Fig. 4). Lineage-specific adaptations are evident—for example, tundra-associated *Nitrospiria* with low-temperature and low-oxygen tolerance[21,27] and desert *Bacilli* with drought resistance[28]. Recent isolation-based studies further confirmed the conservation of cobamide metabolism within specific taxa[16], underscoring the value of taxonomy-informed predictions for guiding future experimental work.

### Selective colonization of cobamide-producing microbes by soil fauna across trophic levels
Soil microorganisms engage in symbiotic, predatory, and competitive relationships with soil fauna, supporting Earth's biodiversity[29–32]. A substantial fraction (25–80%) of these microbes can colonize animal guts[33], providing essential nutrients such as Vitamin B$_{12}$, on which many faunal species are highly dependent (Supplementary Fig. 1). To examine cobamide-producing microbes within soil food webs, we surveyed the gut microbiomes of 60 faunal species spanning trophic levels, climate zones, and land-use types across China (Fig. 3a). Nearly all the gut samples ($n = 664$) from the 60 faunal species were annotated as cobamide-producing microbes via the SCP v.1.0_16S database, from which 230 taxa were identified—classified as very likely (51.3%), likely (9.0%), or possibly (39.7%) producers, predominantly within *Proteobacteria* (59.0%) (Supplementary Fig. 5a, b and Supplementary Data 4). Most taxa presented anaerobic cobamide biosynthesis potential (Supplementary Fig. 5c), which is consistent with stable colonization of the gut environment[34].

Community richness and composition varied markedly among host species but were not clearly associated with land use or geography (Supplementary Fig. 6 and Fig. 3a). Principal component analysis revealed pronounced compositional differences among the six faunal groups (ANOSIM, $R^2 = 0.51$, $p = 0.001$; Fig. 3b and Supplementary Table 1), with PC1 and PC2 together explaining more than 55% of the variation (Fig. 3c). These patterns indicate that host species identity is the dominant factor shaping the gut-associated cobamide-producing microbiome[35]. Interestingly, alpha diversity (Shannon and Chao1 indices) was relatively high in the top-predator species (Kruskal–Wallis test, Shannon: $H(5) = 37.96$, $p < 0.0001$; Chao1: $H(5) = 227.90$, $p < 0.0037$; Fig. 3d), likely reflecting their generalist feeding strategies and the acquisition of cobamide-producing microbes from diverse prey and environmental reservoirs[36]. However, faunal hosts appear to exert selective filtering on microbial transmission[37,38], for instance, via predation across trophic levels

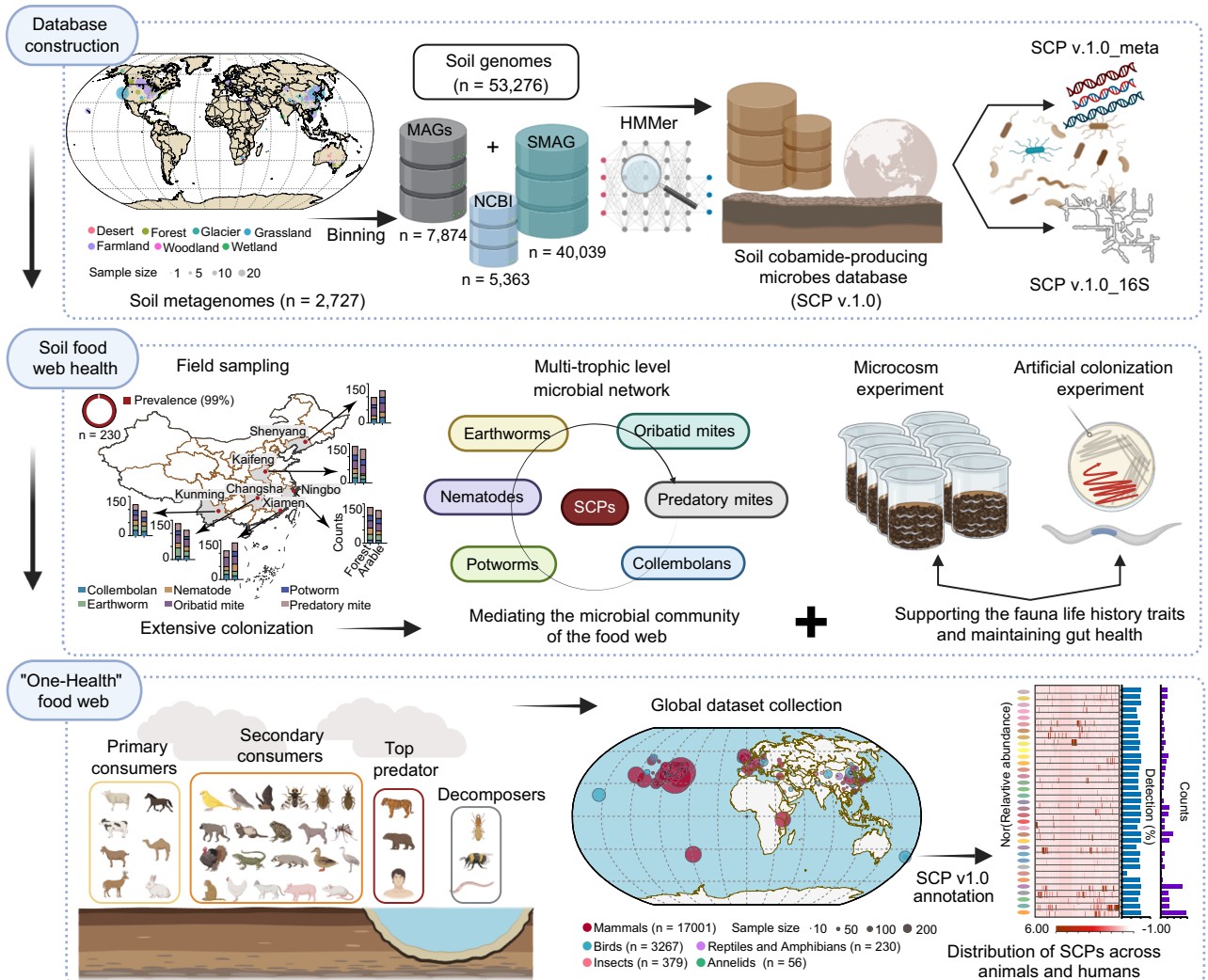

**Fig. 1 | Conceptual framework of this study.** Our study systematically explores cobamide-producing microbes across three interconnected tiers: (1) We first constructed a comprehensive Soil Cobamide Producer (SCP v.1.0) database based on 53,276 soil genomes and characterized their phylogenetic diversity, genomic features, and habitat distributions; (2) we then integrated large-scale field sampling with soil microcosm experiments to examine their functional dynamics across multiple trophic levels, followed by artificial colonization experiments to test their contributions to host nutrition; and (3) finally, we placed these findings in a broader food web context, linking cobamide-producing microbes to higher trophic levels and cross-ecosystem microbial dissemination. MAG metagenome-assembled genomes, NCBI National Center for Biotechnology Information, SMAG soil metagenome-assembled genomes. This figure was created in BioRender. Lu, T. (2025) https://BioRender.com/h35mhr4.

(Fig. 3e). For example, *Xanthomonadaceae* were abundant in collembolans but sharply reduced in predatory and oribatid mites. Such selective filtering was not correlated with the cobamide production potential (Supplementary Fig. 7), implying that additional functional traits influence successful colonization.

## Multifunctional and network hubs of cobamide-producing microbes in soil food webs

Cobamide biosynthesis frequently occurs with other metabolic processes, including amino acid biosynthesis, the production of digestive enzymes, the synthesis of other B vitamins, and the generation of short-chain fatty acids that support gut health (Supplementary Fig. 8). These findings indicate that cobamide-producing microbes are not a single-function group but rather a phylogenetically and metabolically diverse guild in which cobamide biosynthesis represents one of several ecologically relevant traits. Interestingly, the soil fauna appeared to preferentially harbor cobamide-producing taxa whose functional profiles matched their feeding strategies. Detritivores and decomposers, such as collembolans and earthworms, were enriched in

cobamide-producing microbes carrying genes for amino acid and digestive enzyme synthesis (Fig. 3f), facilitating the breakdown of complex organic matter[39]. In contrast, predatory or parasitic mites host cobamide-producing microbes with broader nutrient acquisition repertoires[40]. These host-specific patterns suggest that cobamide biosynthesis may serve as a useful marker for identifying microbial partners with complementary metabolic capacities across different ecological contexts.

To evaluate the broader network role of cobamide-producing taxa, we constructed a multitrophic microbial co-occurrence network from the gut microbiomes of all the sampled fauna (excluding rare taxa; Pearson's $|r| > 0$, $p < 0.05$; Supplementary Fig. 9a). Most likely, producers dominated taxa spanning five to six trophic levels (groups V–VI), whereas nonproducers predominated in groups II–III, indicating that the cobamide biosynthetic trait is more prevalent among microbial taxa linked to higher-trophic-level fauna. Network topology analyses (Supplementary Fig. 9b) identified very likely producers as structural hubs, with significantly higher degrees (median 691 vs. 394 for nonproducers) and closeness centrality values (0.71 vs. 0.60),

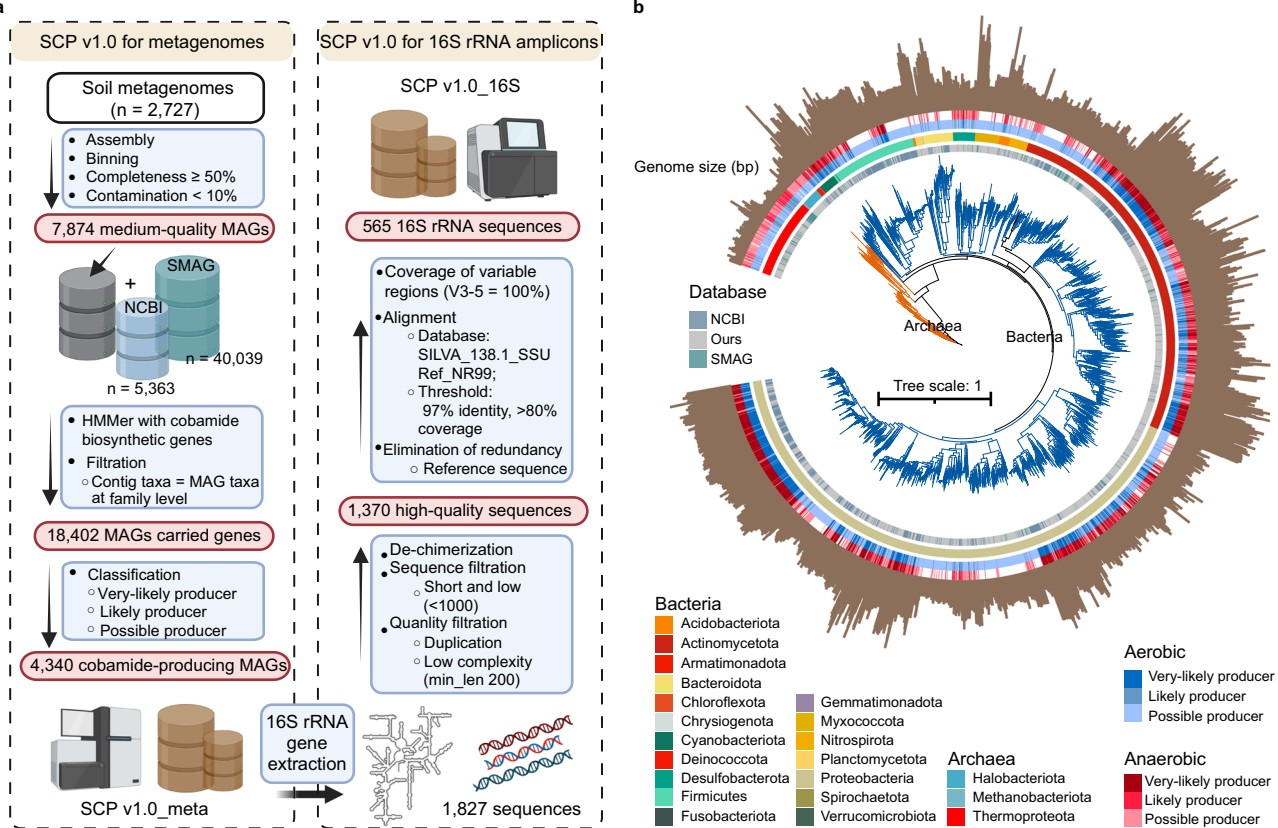

**Fig. 2 | Identification of cobamide-producing microbes across the global terrestrial ecosystem. a** We developed a framework to identify potential cobamide-producing microbes on the basis of metagenome-assembled genomes (MAGs), which were classified as "very likely," "likely," or "possible" producers (see "Methods" for details). On the basis of these results, we constructed the soil cobamide-producer (SCP; v1.0) database for the annotation of metagenomes (SCP v.1.0_meta comprising 4,340 high-quality genomic sequences) and 16S rRNA amplicons (SCP v.1.0_16S comprising 565 high-quality nonredundant sequences). This figure was created in BioRender. Lu, T. (2025) https://BioRender.com/1zbzhsj. **b** Phylogenetic tree of the 4340 cobamide-producing MAGs in soil. Annular heatmaps from inside to outside indicate the database sources, taxonomy, and anaerobic and aerobic classification. The brown column represents the genome size of each MAG. NCBI National Center for Biotechnology Information.

implying greater connectivity and integration within the network. This high centrality may reflect their potential to mediate multinutrient exchanges, including but not limited to cobamides, across hosts at different trophic levels. In contrast, betweenness centrality did not differ significantly, suggesting that nutrient provisioning tends to occur through direct interactions rather than via intermediaries.

Together, these results indicate that while cobamide biosynthesis is not uniquely responsible for soil food web stability, it provides a tractable functional trait for tracing microbial contributions to nutrient exchange networks across trophic levels, in line with a broader "trophic cascading symbiosis" framework[41].

### Dynamic patterns of soil-derived microbes with cobamide biosynthetic traits in *E. crypticus*

The dynamic colonization process by which soil fauna acquire specific microbial taxa from the soil and integrate them into its gut community is central to understanding host–microbe interactions. To test this hypothesis, we performed a soil microcosm experiment in which the model annelid *Enchytraeus crypticus* was used to introduce domesticated individuals with low-diversity gut microbiomes (Supplementary Fig. 10) into natural soil under controlled conditions. Temporal gut microbiome profiles obtained by 16S rRNA amplicon sequencing revealed a deterministic assembly pattern (Supplementary Fig. 11a), indicating selective acquisition from the surrounding soil environment. Principal coordinate analysis revealed three distinct community phases—initial (day 0), dynamic (days 2–7), and stable (days 14–21) (PERMANOVA, $R^2 = 0.61$, $p = 0.001$; Supplementary Fig. 11b, c). Initially,

dominated by *Proteobacteria* (99.8%), the gut community incorporated increasing proportions of soil-derived taxa during the dynamic phase, with *Firmicutes* occupying more stable niches during the stabilization phase. By day 7, soil-derived taxa accounted for $81.1 \pm 5.5\%$ of the gut microbiome (Supplementary Fig. 11d). Among these soil-derived colonizers, only 1.4% presented cobamide biosynthesis potential, indicating that this is a relatively rare but traceable metabolic trait. Cobamide-biosynthetic taxa, such as *Bacillaceae*, *Clostridiaceae*, and *Propionibacteriaceae*, appeared as early as day 2 (Fig. 4a and Supplementary Fig. 12). Their presence shifted over time, with certain operational taxonomic units (OTUs) persisting (e.g., OTU3644), whereas others were transient (e.g., OTU8470, OTU5469, and OTU7054), reflecting host-driven filtering for a subset of microbial partners aligned with metabolic needs.

Stable colonization was assessed across the experimental period, and 119 soil-derived OTUs that consistently persisted in the gut were identified. These OTUs showed significant concordance with host transcriptome variation (Procrustes sum of squares $M^2 = 0.2082$, $r = 0.5094$, $p < 0.001$; Supplementary Fig. 13a), suggesting their importance in shaping host functional capacity (Supplementary Fig. 13b). Among these, seven taxa with cobamide biosynthetic potential maintained stable colonization (Fig. 4b, c). Notably, *Bacillus megaterium* (OTU3644) and *Streptomyces* sp. 3485 (OTU359) were strongly correlated with host gene expression profiles (Pearson's $r > 0.6$, $p < 0.05$; Fig. 4d and Supplementary Fig. 14). The associated differentially expressed genes mapped to several metabolic pathways (Fig. 4e), such as folate metabolism, whereas in soil, fuana cobamide-

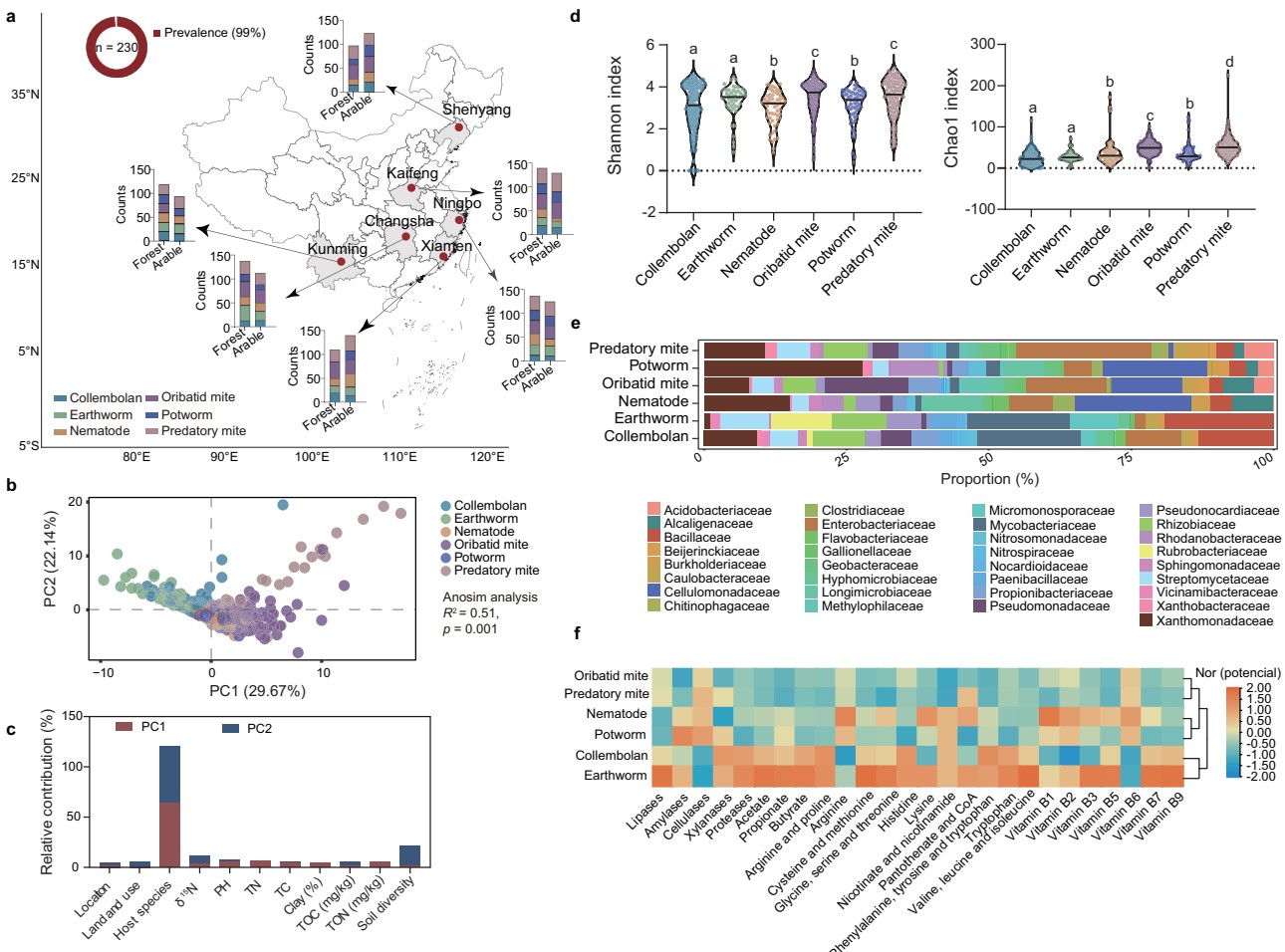

**Fig. 3 | Selective colonization of cobamide-producing microbes by soil fauna across trophic levels. a** Geographic distribution of cobamide-producing microbes in the guts of various soil fauna across different regions in China, with sampling locations marked on the map. Each sampling location indicates the number of cobamide-producing OTUs detected in the guts of collembolans, oribatid mites, earthworms, potworms, nematodes, and predatory mites. Most of the soil fauna (99% of the samples) harbored cobamide-producing microbes. **b** Principal component analysis further revealed distinct patterns of cobamide-producing microbes among the six species groups according to analysis of similarity (ANO-SIM) ($R^2 = 0.51$, $p = 0.001$). **c** The relative contributions of various environmental factors to the variation in cobamide-producing microbes in the gut. Principal component 1 (PC1) and PC2 represent the main axes of variation, with significant contributions from location, land use, soil pH, and other factors. TOC total organic carbon, TN total nitrogen, TON total organic nitrogen. **d** Alpha diversity indices (Shannon and Chao1) were used to compare the microbial diversity in the guts of different soil fauna. Lowercase letters above each violin plot denote significant differences between groups. **e** Taxonomic composition of cobamide-producing bacterial families in the guts of various soil fauna. The proportions of bacterial families are shown, highlighting differences in microbial community structure among host species. **f** Heatmap displaying the functional potential (normalized value) of cobamide-producing microbes in the gut of soil fauna, with a focus on enzyme pathways associated with vitamins, short-chain fatty acids, and amino acid metabolism. The color intensity represents the relative potential of each function, with clustering on the basis of similarity. Source data are provided as a Source Data file.

dependent methionine synthase links Vitamin B$_{12}$ availability to the folate cycle.[42] However, these taxa are also linked to other functional capacities, such as the biosynthesis of digestive enzymes, other B vitamins, and amino acids, underscoring the multifunctional nature of colonizing taxa and supporting the view that nutrient-driven host–microbe interactions are multifaceted, extending beyond reliance on a single compound.

## Host functional impacts of a cobamide-producing bacterium in *E. crypticus*

Two gut-colonizing cobamide-producing strains of *E. crypticus*, *Streptomyces violaceus* NBC_00450 (OTU359) and *Bacillus megaterium* ATCC 14581 (OTU3644), were identified via BLAST analysis of the NCBI Microbial Genomes database (99% identity and 100% coverage). Genome analysis confirmed that both strains possessed complete genetic potential for cobamide biosynthesis (Supplementary Data 5), and direct measurements verified Vitamin B$_{12}$ production (Fig. 5a). At

equivalent cell densities ($10^5$ CFU/mL), *B. megaterium* produced 6.8-fold more Vitamin B$_{12}$ than did *S. violaceus* (two-sided *t* test, t(4) = 5.57, $p < 0.05$, 95% CI [−1.24, −0.41]). This difference was reflected in the results of the supplementation assays, which revealed that 7 days of oral administration of *B. megaterium* significantly increased certain growth parameters in *E. crypticus*, whereas *S. violaceus* supplementation had no detectable effect compared with that in the control group (two-sided *t* test, statistical values see Supplementary Data 6 and Fig. 5b).

Given this, we focused subsequent colonization assays on *B. megaterium* ATCC 14581, a previously characterized soil-derived isolate[43]. Preliminary tests established $10^5$ CFU/mL as an optimal inoculation dose that did not negatively affect survival (100%) or oviposition rates (Supplementary Fig. 15). Supplementation with *B. megaterium* or with Vitamin B$_{12}$ at equivalent concentrations (approximately 1 µg/L) significantly increased egg hatching rates and larval length (two-sided *t* test, statistical values see Supplementary

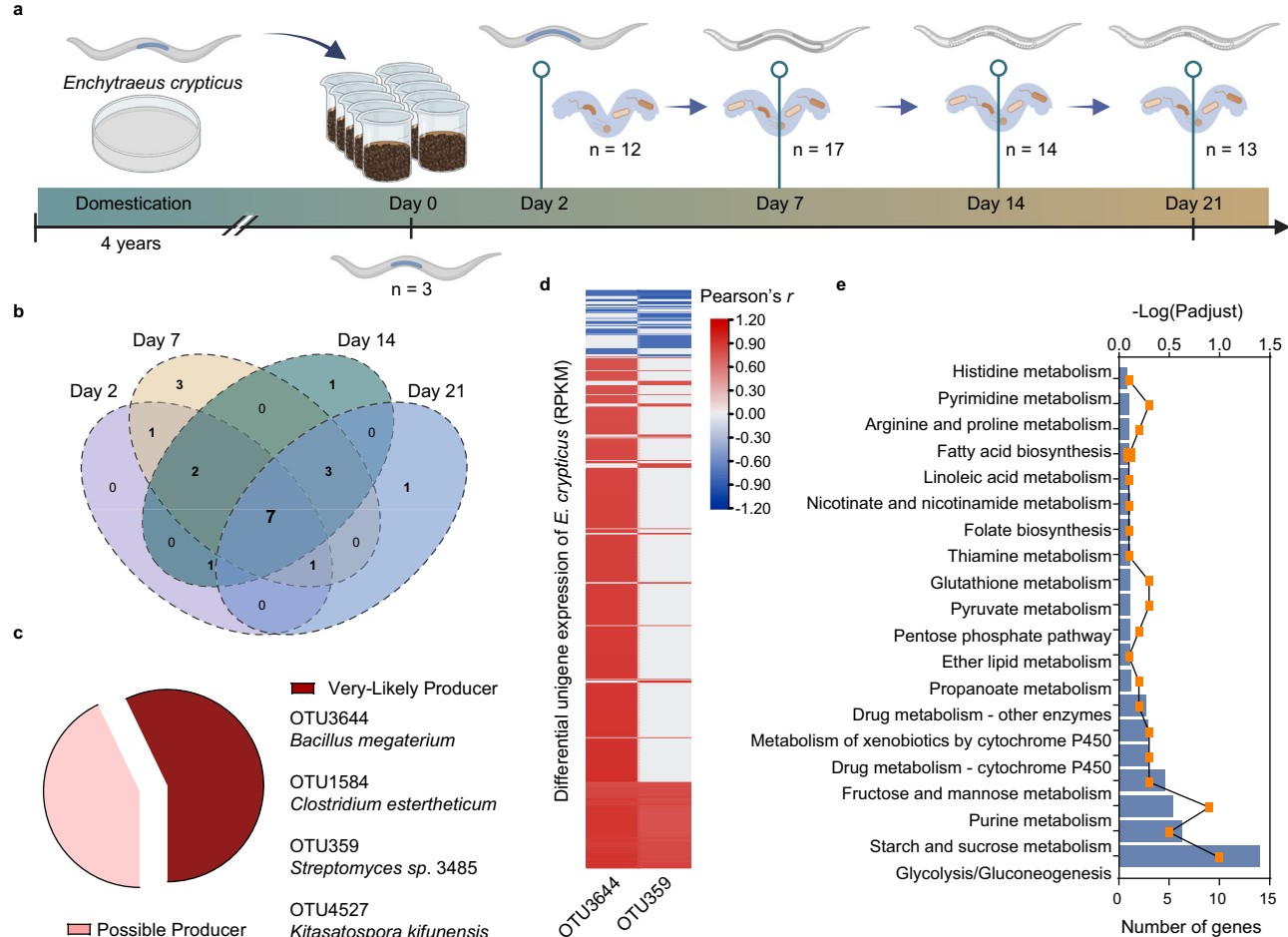

**Fig. 4 | Dynamic patterns of soil-derived microbes with cobamide biosynthetic traits in *Enchytraeus crypticus*.** **a** Schematic representation of a microcosm experiment conducted to study the temporal dynamics of cobamide-producing microbes within the gut of *E. crypticus* over 21 days. A total of 3, 12, 17, 14, and 13 cobamide-producing OTUs were detected in the gut of *E. crypticus* on days 0, 2, 7, 14, and 21, respectively. This figure was created in BioRender. Lu, T. (2025) https://BioRender.com/8b6dyqg. **b** Venn diagram showing the shared and unique cobamide-producing OTUs present in the gut of *E. crypticus* across different sampling days. Seven cobamide-producing microbes were consistently detected throughout the experiment. **c** Pie chart showing the classification of cobamide-producing OTUs identified in the *E. crypticus* gut. Four of these producers, OTU3644 (*Bacillus megaterium*), OTU1584 (*Clostridium esterth-eticum*), OTU359 (*Streptomyces* sp. 3485), and OTU4527 (*Kitasatospora kifunensis*), are classified as

"very likely" producers on the basis of their predicted biosynthetic capability. **d** Heatmap displaying Pearson's correlation (*r*) between the gene expression pro-files of *E. crypticus* and the abundance of OTU3644 (*Bacillus megaterium*). Strongly correlated genes (|r| ≥ 0.7, adjusted *p* < 0.05) are highlighted, with red and blue indicating positive and negative correlations, respectively. Statistical significance was assessed using two-sided Pearson correlation analysis. **e** The genes sig-nificantly correlated with *Bacillus megaterium* were predominantly involved in glycolysis and other beneficial functions. The yellow dots represent the sig-nificance of gene enrichment (−Log(*P*-adjust)), whereas the blue columns indicate the corresponding number of genes. Functional enrichment significance was determined using a two-sided Fisher's exact test, and *p*-values were adjusted for multiple comparisons using the Benjamini-Hochberg method. Source data are provided as a Source Data file. OTU operational taxonomic unit.

Data 6 and Fig. 5c, d), accelerated larval development and enhanced adult oviposition (two-sided *t* test, statistical values see Supplementary Data 6 and Fig. 5e, f). These effects are consistent with prior reports on other soil invertebrates[11], suggesting that cobamide production is a key contributor, although other metabolites from *B. megaterium* may also play a role.

To assess the specific role of cobamides, we modulated Vitamin $B_{12}$ production in *B. megaterium* by altering cobalt ion availability in growth medium[44] (one-way ANOVA, $F_{(2, 27)} = 25.08$, $\eta^2 = 0.65$, $p = 0.0037$; Fig. 5g). Host growth responses were positively correlated with bacterial Vitamin $B_{12}$ production (two-sided one-way ANOVA, $F_{(2, 27)} = 16.31$, $\eta^2 = 0.88$, $p < 0.0001$; Fig. 5h), indicating that this trait can be a measurable determinant of certain life-history traits in *E. crypticus*. Interestingly, *B. megaterium* supplementation also led to community-level shifts; compared with Vitamin $B_{12}$ alone, *B. megaterium* supple-mentation increased the relative abundance of taxa such as

*Actinobacteria* and *Bacteroides* and elevated overall gut microbial diversity (Supplementary Fig. 16). Functional profiling revealed enrichment of pathways associated with host immunity and growth, including folate biosynthesis (ko00790), biotin metabolism (ko00780), the NOD-like receptor signaling pathway (ko04621), and flagellar assembly (ko02040), in the *B. megaterium* group relative to both the control and Vitamin $B_{12}$ groups (Fig. 5i). These findings indicate that cobamide biosynthesis is a representative microbial trait rather than a unique driver.

## Global distribution of cobamide-producing microbes across diverse host taxa

Cobamide-producing microbes not only provide hosts with essential micronutrients but also contribute additional functions that may support host growth, metabolism, and overall physiological perfor-mance. Their multifunctional traits make them relevant partners in

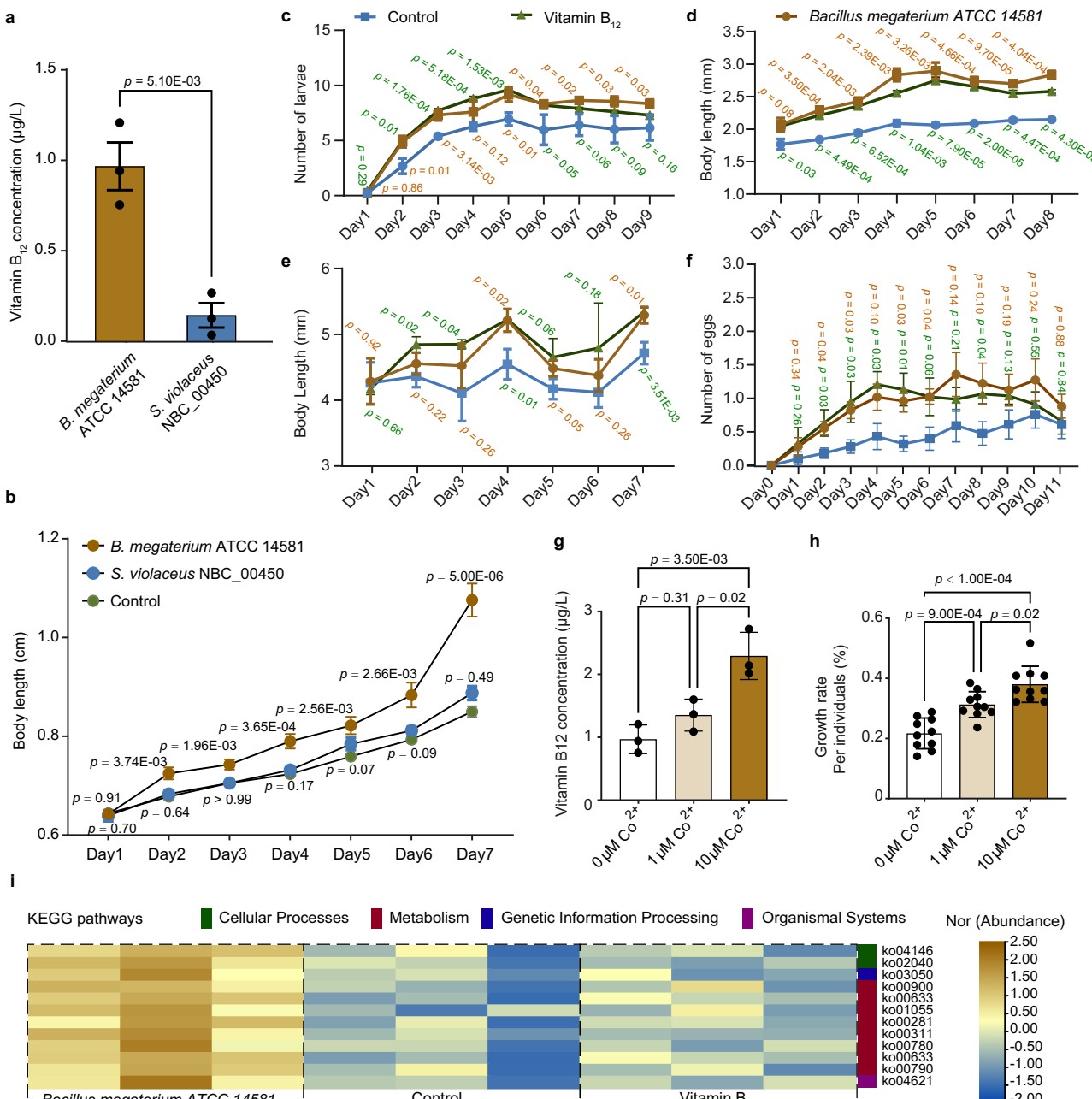

**Fig. 5 | Host functional impacts of a cobamide-producing bacterium in *E. crypticus*. a** Vitamin B$_{12}$ production by *Bacillus megaterium* ATCC 14581 and *Streptomyces violaceus* strain NBC_00450 at 10$^8$ CFU/mL (*n* = 3 biological replicates per strain). Statistical significance was assessed using a two-sided *t* test (*t*(4) = 5.568, *p* = 0.0025, 95% CI = [−1.24, −0.41]). **b** Length of *E. crypticus* adults after oral supplementation with *B. megaterium* ATCC 14581 or the *S. violaceus* strain NBC_00450 (*n* = 10 biological replicates per treatment). Statistical significance was determined using two-sided unpaired *t* test (between control and treatment). **c** Hatchability of *E. crypticus* eggs (*n* = 10), **d** length of *E. crypticus* larvae (*n* = 10), **e** length of *E. crypticus* adults (*n* = 10), and **f** ovulation rate (*n* = 10) after oral supplementation with *B. megaterium* ATCC 14581 (10$^5$ CFU/mL) and Vitamin B$_{12}$ (positive control). The blue, brown and green dotted lines indicate the control, *B. megaterium* ATCC 14581 and Vitamin B$_{12}$ treatment groups, respectively. Statistical significance was determined using a two-sided unpaired *t*-test (between control and treatment). Exact statistical data for panels b-f are provided in Supplementary Data 6. **g** Vitamin

B$_{12}$ concentrations detected in *Bacillus megaterium* ATCC 14581 cultures supplemented with 0, 1, or 10 μM CoCl$_2$ after 48 h (*n* = 10). Data were analyzed by two-sided one-way ANOVA (*F*(2, 27) = 25.08, η$^2$ = 0.65, *p* = 0.0037) with Tukey's multiple comparison test. **h** Growth rates of adult *E. crypticus* on day 7 following oral administration of *Bacillus megaterium* ATCC 14581 supplemented with different concentrations of CoCl$_2$ (0, 1, and 10 μM) (*n* = 10). Statistical significance was determined using a two-sided one-way ANOVA [*F*(2, 27) = 16.31, η$^2$ = 0.88, *p* < 0.0001] with Tukey's multiple comparison test. **i** Oral supplementation with *B. megaterium* ATCC 14581 significantly enhanced the functions of the gut microbiome, including cellular processes (green), metabolism (red), genetic information processing (blue) and organismal systems (purple). The color scale represents the normalized abundances from high (brown) to low (blue). Data are presented as mean ± s.e.m. from independent biological replicates (each replicate derived from a separate batch of individuals or cultures). No technical replicates were used for statistical inference. Source data are provided as a Source Data file.

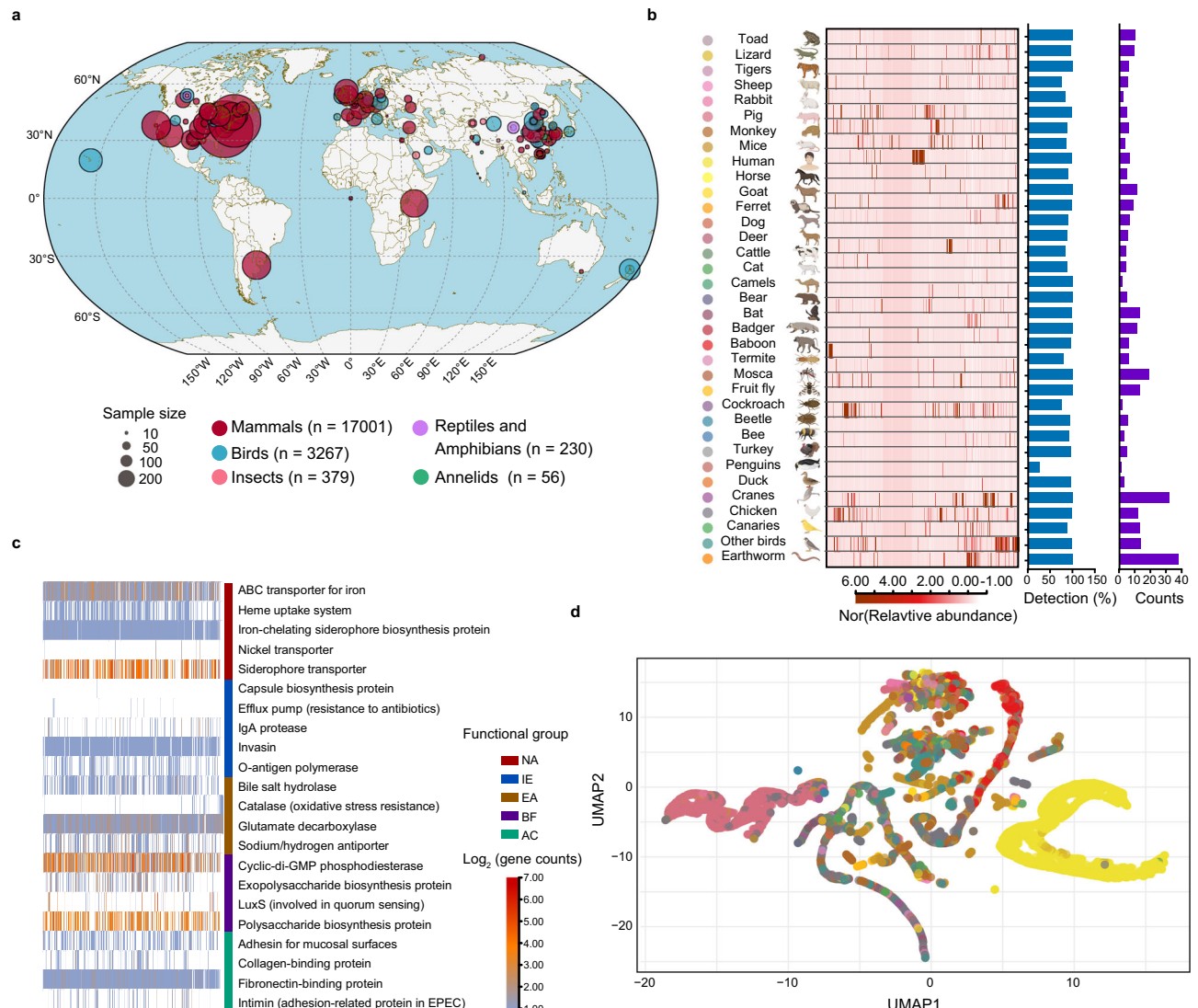

**Fig. 6 | Global distribution of cobamide-producing microbes across diverse host taxa. a** We collected 20,932 16S rRNA amplicon sequencing data from the gut microbiomes of 19 mammals (including humans), seven birds, seven insects, one annelid, and two amphibian and reptile species across 150 locations and 31 continents. **b** Cobamide-producing microbes were annotated in these amplicon sequencing data on the basis of the SCP v.1.0_16S database. Most humans and animals harbor soil cobamide-producing microbes in the gut microbiome. The red color scale indicates the normalized relative abundance from low (light) to high (dark). **c** The functional potential of cobamide-producing microbes associated with gut colonization includes nutrient acquisition, immune evasion, environmental adaptation, biofilm formation, and adhesion and colonization. The color scale represents normalized counts per genome, ranging from low (blue) to medium (orange) and high (red). **d** Distribution of cobamide-producing microbes in the UMAP dimensionality reduction space. Different colors indicate different animal groups. The intercluster distances in the UMAP plots could not be interpreted quantitatively. UMAP uniform manifold approximation and projection, NA nutrient acquisition, IE immune evasion, EA environmental adaptation, BF biofilm formation, AC adhesion and colonization. Source data are provided as a Source Data file.

maintaining the health and ecological balance of both humans and animals. To explore the extent to which these producers from soil food webs occur in animal and human gut systems, we analyzed their distribution across diverse host taxa.

From 20,932 16S rRNA amplicon sequences obtained at 150 locations in 31 countries across six continents, we annotated 471 OTUs as putative cobamide-producing microbes (Fig. 6a and Supplementary Data 7). These taxa were detected in the guts of 19 mammalian species (including humans), seven bird species, seven insect species, one annelid, and two amphibian/reptile species, indicating that cobamide-producing microbes are widely distributed across phylogenetically diverse hosts (Fig. 6b) owing to their ability to colonize and adapt to the gut environment (Fig. 6c) through key functions such as nutrient acquisition, immune evasion, environmental adaptation, biofilm formation, and adhesion. To test whether colonization patterns were

determined by host exposure to soil environments, we quantified each animal species' degree of soil contact via five ecological indicators (activity range, diet, activity time/frequency, dust exposure, and food web associations) (Supplementary Data 8). However, no significant correlation was detected between soil contact and cobamide-producing microbe colonization (ordinary least squares regression, $P > 0.05$; Supplementary Fig. 17), suggesting that host-specific factors, rather than environmental exposure alone, may shape these associations.

Community-level analyses confirmed species-specific patterns. ANOSIM revealed significant compositional differences among the animal groups ($R^2 = 0.49$, $p = 0.01$), and UMAP visualization revealed consistent clustering by host taxa (Fig. 6d), albeit without quantitative interpretability due to nonlinear projection. Notably, cobamide-producing microbes, particularly omnivorous species such as cranes that consume soil fauna, are relatively enriched in the guts of birds,

suggesting a potential route of transmission from soil food webs to vertebrate hosts. Many herbivorous birds, which obtain limited Vitamin $B_{12}$ from plant-based diets, likely rely on gut microbes to supplement this essential nutrient[45]. Interestingly, insects, which generally lack a direct Vitamin $B_{12}$ requirement, also harbor abundant cobamide-producing microbes, possibly reflecting their involvement in mutualistic or syntrophic interactions with other microbes. These findings emphasize that the ecological roles of cobamide-producing microbes extend beyond simple micronutrient supply and may differ substantially across animal groups.

In summary, this study positions cobamide-producing microbes as powerful and experimentally tractable models for understanding microbial nutrient provisioning, with implications that extend across ecological and One Health frameworks. Despite their phylogenetic diversity, these microbes consistently colonize animal hosts, modulate gut communities, and promote host development through both Vitamin $B_{12}$ biosynthesis and broader metabolic contributions. By integrating a global database with ecological analyses and host interaction experiments, we demonstrate how microbial nutrient provisioning underpins trophic stability and ecosystem health. More broadly, we propose that cobamide-producing microbes exemplify a general principle of nutrient dependency that may extend across diverse ecological and host-associated systems. These insights underscore the multifunctionality of microbes in sustaining ecological balance and open new avenues for One Health strategies that integrate environmental, animal, and human well-being.

## Methods
Detailed bioinformatics analysis, experimental procedures, additional materials, and technical protocols are provided in the Supplementary Materials and Methods.

### Metadata analysis
We compiled four datasets: representative soil animal genomes (Supplementary Data 9), large-scale soil metagenomes, representative bacterial genomes (Supplementary Data 10), and 16S rRNA amplicon data. A total of 132 soil invertebrate genomes (19 orders) were retrieved from NCBI (June 2023), and the most complete and recent assemblies with available annotations were selected and quality checked with BUSCO (>95% completeness). From ENA (January 2022), we obtained 2727 soil metagenomes from 24 countries, excluding plant-associated, non-Illumina, and low-quality data, alongside 5363 soil bacterial genomes from NCBI and 40,039 high-quality MAGs from the SMAG catalog. To assess cobamide-producing microbes across host microbiomes, we collected 20,932 gut 16S datasets from 19 mammals, 7 birds, 7 insects, 1 annelid, and 2 amphibian/reptile species across 31 countries, excluding nongut and intervention-related samples. Metagenomic reads were filtered with Trimmomatic, assembled via MEGAHIT, and binned with CONCOCT, MaxBin2, and metaBAT2; CheckM identified 7874 medium-quality MAGs (≥50% completeness, <10% contamination). Cobamide biosynthesis genes were detected via HMM profiles from KEGG, TIGRFAM, PFAM, and UniProt (HMMsearch, $E \le 1e{-}6$), and MAGs were classified into four phenotypes (very likely, likely, possible, and nonproducer) (Supplementary Data 11). After contig-level filtering, 4,340 producer MAGs were retained, dereplicated into 1,853 species-level MAGs with dRep, and functionally annotated to establish the SCP (v.1.0_meta) database. From the producer MAGs, 1827 16S rRNA sequences were extracted and curated into 565 high-quality nonredundant sequences (SCP v.1.0_16S), while 2379 sequences from nonproducer MAGs formed a control set, and amplicon-based identification of producers/nonproducers was performed via BLASTn with 100% identity.

### Field experimental design and data analysis
Field sampling was conducted across six sites in China (24.9-41.7°N, 102.95–123.72°E) between October and November 2017, covering major climatic zones, soil types, and vegetation conditions to capture ecological diversity (Supplementary Data 12). In total, 238 collembolan (~3600 individuals), 60 nematode (approximately 6000), 62 potworm (approximately 1500), 146 oribatid mite (approximately 4000), 122 predatory mite (approximately 2000), 50 earthworm (approximately 1000), and 60 bulk soil samples were collected for microbial and isotopic analyses. The target taxa represented >75% of the global soil faunal biomass and spanned four trophic levels: decomposers (earthworms, potworms), microbivores (nematodes, collembolans), secondary consumers (oribatid mites), and predators (predatory mites). Dominant fauna were identified morphologically and via DNA barcoding. Individuals were surface sterilized and homogenized, and DNA was extracted via the DNeasy Blood and Tissue Kit (QIAGEN, Germany), with yields normalized to $1\,ng\,\mu L^{-1}$ prior to PCR and stored at $-20\,°C$ for downstream analyses.

### Laboratory experimental design
The test soil was sandy loam (clay (2.27%), silt (16.47%), and sand (81.25%)) collected from the topsoil layer (0–20 cm) of arable land sites (29° 49′N, 121° 20′E; Zhejiang, China). The soil was air-dried in the shade, cleared of root debris, pebbles, and other allogenic materials, and then filtered through a 2 mm (pore size) plastic soil sieve. The following basic physicochemical properties of the soil samples were obtained following previous protocols[33]: pH, 5.16; water-holding capacity, 46.8%; and total nitrogen concentration, $3.8\,g\,kg^{-1}$. E. crypticus (originally obtained from Aarhus University, Denmark) was cultivated in our laboratory for 5 years following the Organization for Economic Cooperation and Development (OECD) guideline 220 (OECD, 2004) and maintained in Petri dishes covered with culture medium[46]. Bacillus megaterium ATCC 14581 and Streptomyces violaceus NBC_00450 (purchased from BeNa Culture Collection, Beijing, China) were grown in culture media ($CaCl_2 \cdot 2H_2O$, $MgSO_4$, KCl, $NaHCO_3$, and Bacti-Agar media [Agar No. 1; Oxoid]). Vitamin $B_{12}$ (cyanocobalamin, CAS: 68-19-9, USP Grade, Assay 99%) was purchased from Sangon Biotech (Shanghai, China) and mixed with water to prepare a 1 mg/L stock solution.

All E. crypticus and eggs of the same age were obtained synchronously according to the Organization for Economic Cooperation and Development (OECD) guidelines (OECD, 2004). We transferred E. crypticus individuals of similar body size and capable of spawning to new, clean Petri dishes, which were marked with laying dates. Fresh eggs (<24 h) were removed, introduced into a new Petri dish, and labeled synchronized eggs. Hatched larvae (<24 h) were defined as synchronized E. crypticus. Soil microcosms (90 g of soil in glass beakers, $n = 16$) were inoculated with 30 synchronized adults and incubated under controlled conditions for up to 21 days; soil and gut samples were collected at days 0, 2, 7, 14, and 21 for DNA/RNA analyses. To test the effects of bacteria on host growth, synchronized adults or eggs were exposed to B. megaterium or S. violaceus suspensions ($10^4–10^8\,CFU\,mL^{-1}$), Vitamin $B_{12}$, or sterile water controls. Additional assays have examined bacterial dose–response, host development, reproduction, and the influence of $Co^{2+}$ supplementation on B. megaterium cobamide production and host growth. The vitamin $B_{12}$ levels in the bacterial lysates were quantified via an ELISA kit (Shanghai Tongwei Biotechnology).

### 16S rRNA gene amplicon sequencing and bioinformatics analysis
The V4 hypervariable region of the 16S rRNA gene was amplified via universal primers (515 F: forward GTGCCAGCMGCCGCGGTAA and 806 R: reverse GGACTACHVGGGTWTCTAAT), each tagged with unique barcodes to distinguish samples. PCR reactions (25 μL) contained 12.5 μL of 2× high-fidelity PCR master mix, 0.2 μM of each primer, and approximately 10 ng of template DNA. Thermal cycling conditions were 95 °C for 3 min, followed by 25 cycles of 95 °C for 30 s,

55 °C for 30 s, and 72 °C for 30 s, with a final extension at 72 °C for 5 min[33]. Amplicons were purified, quantified, pooled, and sequenced on an Illumina MiSeq platform (Novogene, Beijing, China). The bacterial 16S rRNA gene sequences were analyzed via Quantitative Insights into Microbial Ecology (v.1.9.1) following the platform's online guidelines[46]. Briefly, the raw sequences were filtered to retain only high-quality reads by removing barcodes, low-quality reads, and ambiguous bases. The sequences were clustered into OTUs with 97% similarity via an OTU picking approach. To improve data accuracy, singletons were discarded, and representative sequences for each OTU were aligned via PyNAST[47]. Taxonomic classification of the OTUs was performed via the SILVA (v.138) SSU reference database with the RDP Classifier (v.2.2)[48]. To minimize sequencing noise, OTUs with <100 reads across all samples were excluded from the analysis.

### RNA extraction and transcript sequencing
A set of 20 adult *E. crypticus* per replicate was flash-frozen in liquid nitrogen and stored at −80 °C to maintain RNA integrity. RNA was isolated from each replicate pool via the HiPure Universal RNA Midi Kit (Magen, Guangzhou, China). The concentration and purity of the RNA were assessed via a Nanodrop 2000 spectrophotometer, and agarose gel electrophoresis was performed to verify RNA integrity. The Agilent 2100 Bioanalyzer system was used to measure RNA integrity to ensure that samples met the following criteria for library preparation: total RNA ≥ 1 μg, concentration ≥ 35 ng/μL, OD260/280 ≥ 1.8, and OD260/230 ≥ 1.0. Oligomagnetic beads were used to isolate mRNA from the total RNA. The enriched mRNA was then randomly fragmented to approximately 300 bp and reverse transcribed to generate stable double-stranded cDNA via the ReverTra Ace qPCR RT Kit (TOYOBO, Osaka, Japan), which was then sequenced on an Illumina NovaSeq 6000 platform (Meiji, Shanghai, China).

### Statistical analysis and visualization
The data are presented as the means ± standard errors of the means. The Kolmogorov–Smirnov test was used to assess the normality of the data distribution. The Brown–Forsythe test and F test were conducted to examine the homogeneity of variances for two groups or more, respectively. When the data were normally distributed and the variances were homogeneous, one-way analysis of variance (ANOVA) with Bartlett's test was used for multiple-group comparisons, and Student's one-sided *t* test was employed for between-group comparisons. If the data did not meet these assumptions, the Mann–Whitney and Kruskal–Wallis tests were applied for pairwise comparisons of two or more groups. Power analyses for *t* tests and ANOVA were conducted to ensure adequate sample sizes for detecting significant effects. All the statistical analyses and graphical visualizations were performed via GraphPad Prism 9 and IBM SPSS Statistics (v.20.0.0). Principal coordinate analyses and Adonis tests were performed via the "vegan" (v.2.6-2) and "pairwiseAdonis" (v.0.4.1) packages in R (v.3.6.3), respectively, and heatmaps were generated via TBtools (v.1.082). Correlations within the relative abundance of the prokaryotic species network were calculated for each group via Pearson's correlation ($P < 0.05$) in R, and the calculation of network topology data was performed via Gephi (v.10.0.3). Venn diagrams were created via EVenn software (http://www.ehbio.com/test/venn). Data figures were constructed via GraphPad Prism 9, and the "ggplot2" (v.3.5.0) package in R (v.3.6.3), and Adobe Illustrator 2020 was used for figure formatting. All schematic diagrams and art elements were designed via BioRender (https://www.biorender.com). Maximum-likelihood phylogenetic trees were constructed via ITOL. Gut microbial function was predicted via PICRUSt2 (www.majorbio.com). Maps used in Figs. 3a, 6a, and Supplementary Figs. 2(a), 3, and 4(a) were generated in Python using the "Basemap" package (v.1.3.6) with publicly available coastline and land-mass data from the Natural Earth dataset (https://www.naturalearthdata.com), which is released under the public domain. The use of Basemap and

Natural Earth data complies with their respective terms of use and requires no additional licensing for publication.

### Reporting summary
Further information on research design is available in the Nature Portfolio Reporting Summary linked to this article.

## Data availability
The raw 16S rRNA gene amplicon sequencing data generated in this study have been deposited in the NCBI Sequence Read Archive under BioProject accession code PRJNA916760. The genomes of *Streptomyces violaceus* NBC_00450 and *Bacillus megaterium* ATCC 14581 are available in public repositories, and their accession numbers are SAMN24016225 and SAMN15857556, respectively. The raw metagenomic datasets generated in this study have been deposited in public repositories, and the corresponding accession numbers are provided in Supplementary Data 2. Accession numbers for the publicly available 16S rRNA amplicon datasets analyzed here are listed in Supplementary Data 7. All soil faunal reference genomes included in this study are publicly accessible, with detailed accession information given in Supplementary Data 9. In addition, the soil bacterial genomes examined in this work are available from public repositories, and their accession numbers are summarized in Supplementary Data 10. The transcriptome dataset generated from *Enchytraeus* in this study has been deposited in the NCBI SRA under the BioProject accession PRJNA1373487. The SMAG catalog generated by Ma et al.[26] is publicly accessible via the Zenodo repository at https://doi.org/10.5281/zenodo.7341719. All other processed data supporting the findings of this study are provided in Supplementary Information. Source data are provided with this paper.

## Code availability
All scripts and codes for machine learning, visualization, bioinformatics, and statistical analyses related to this study are available online at https://github.com/QiZhang11/SCP-as-a-model-for-nutritional-interdependencies-in-soil-food-webs [49].

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

## Acknowledgements

This study was financially supported by the National Natural Science Foundation of China (42307158, 22376187), Zhejiang Provincial Natural Science Foundation of China (LZ23B070001), and the Spanish Government grants PID2022-140808NB-I00 and TED2021-132627 B-I00 funded by MCIN, AEI/10.13039/501100011033 European Union Next Generation EU/PRTR.

## Author contributions

Q.Z. designed the study with guidance from H.F.Q. and D.Z. Q.Z. wrote the first draft of the manuscript and H.F.Q., D.Z., Y.-G.Z., J.P., J.C., and J.Q.S. contributed substantially to revisions. Q.Z. B.F.C., Z.Y.Z. (Zhenyan Zhang) and N.H.X. performed all the metagenomic analysis. Q.Z. and J.C.W. contribute to the identification of cobamide-producing microbes using metagenome-assembled genomes. Q.Z., B.F.C., and T.L. were responsible for the Machine learning model construction and related data analysis. Q.Z. M.K.J., Y.T.Y., Q.P., and Z.Y.Z. (Ziyao Zhang) perform all the experiments. Q.Z. performed the visualization of all data and the

artistic design of all figures. H.F.Q., Q.Z. and J.P. acquired funding for this project.

## Competing interests

The authors declare no competing interests.
