## [Transparent Peer Review file · Nature Communications]

Cobamide-producing microbes as a model for understanding general nutritional interdependencies in soil food webs

Corresponding Author: Professor Haifeng Qian

Version 0:

Reviewer comments:

Reviewer #1

(Remarks to the Author)

The manuscript by Zhang et al titled, "Cobamide producer driving soil food web health" brings together a combination of genomic and experimental evidence to investigate cobamide producers in soil and across the soil food web, and posits that these producers have important roles at each of these trophic levels. There are a number of major issues that permeate this manuscript, including misleading claims, methodological errors, a lack of acknowledgement of previous literature, and a disjointed narrative. Thus, I do not recommend publication in its current form.

Claims throughout the manuscript are misleading. Exemplary of this issue is the title, which claims that cobamide producers drive the process of soil food web health, despite not comparing cobamide producers to non-producers. The authors present evidence that cobamide-producing bacteria are present in the soil and across the guts of different soil fauna, but do not demonstrate that cobamide production capacity is what enables these bacteria to support soil food web health. The authors experimentally show that *Bacillus megaterium*, a cobamide producer, benefits the growth of the model worm species *Enchytraeus crypticus*. However, they fail to link this growth advantage to the cobamide-producing capacity of the *Bacillus* species. To prove this would require a knockout in cobamide-production in *B. megaterium* or another method to alter the total cobamide produced.

There are also issues with the methods. Two illustrative examples: (1) the use of an E-value cutoff of 10^{-6} is a very loose metric for most HMM searches, which often require E-value cutoffs less than 10^{-20} or lower (depending upon the HMM in question) to be trusted. (2) UMAP is a 2-dimensional manifold that clusters by local proximity, but it is non-linear, non-deterministic, and does not preserve global structure, and should not be analyzed using statistical methods. Additionally, some of the methods are not described adequately, for example which steps in the cobamide biosynthetic pathway were used to assign MAGs as producers.

This work does not correctly incorporate or acknowledge work previously published by other groups. For instance, the authors claim to have developed a novel method to classify cobamide producers without explaining this classification process. However, it appears that they classify microbes as "very likely," "likely," and "possible" cobamide producers similarly to the method used by Shelton et al, ISME 2018, and thus is not novel. Furthermore, the database they create is another in a series of online microbial cobamide metabolism-related databases (e.g. Zhou et al, mSystems 2021, Wang et al, ISME 2024). As another illustrative example, the authors assume that 16S sequence is a good proxy for metabolic potential for cobamide production (in contrast to work that challenges this assumption, for instance, Alvarez-Aponte et al. ISME 2024).

Finally, the narrative arc of this work is disjointed; there is no logical flow linking identification of these producers in various fauna gut microbial communities, in vivo worm-based experiments, and analysis of cobamide producers in various environments. Two examples where the narrative is illogical within individual sections: (1) line 233 — the authors conclude that there is a symbiotic supply-demand dynamic between soil fauna and cobamide producers without providing any evidence that the interaction is "symbiotic" or possesses a "supply-demand" in nature. (2) figure 3d — The authors find that one cobamide-producing bacterium (*Bacillus*) has a positive health benefit to *Enchytraeus crypticus*, but ignore the fact that another cobamide-producing bacterium (*Streptomyces* sp. 3485) has no such benefits.

(Remarks on code availability)

Reviewer #2

(Remarks to the Author)

(Remarks on code availability)

Reviewer #3

(Remarks to the Author)

In their manuscript the authors Zhang et al. are reporting a very comprehensive study identifying the occurrence of cobamide producers in existing genome databases, observing their abundance in soil fauna in a comprehensive field study, identifying them in human and animal guts and studying the mechanisms of gut colonization as well as functional importance for the host organism using model organisms. The title "Cobamide producer driving soil food web health" does not nearly frame the complexity of this study. This complexity is further reflected in the 19 Figures in the Supplementary Materials. They further make some key discoveries. Besides identifying cobamide producers in existing soil metagenome databases they find a strong species specificity in host colonization in soil fauna as well as other animals and that higher trophic organisms have increased diverse gut microbiomes compared to lower trophic levels indicating that higher trophic levels ingest and incorporate the microbiome of their prey. Additionally, they show the colonization process of the host gut and can link the presence of cobamide producers to increased fitness and reproduction rates in soil fauna.

The study is presenting a very interesting and significant topic. The role of cobamide producers for host health and consequences for ecological functioning expands to fit the One Health concept. The execution of the genome database usage as well as experimental studies appear sound.

The data presentation would benefit from increased quality figures and most figures are too small. The text needs revision for clarity, grammar and spelling.

I believe this study is eligible for publication, however it would greatly benefit from greater focus and reduction of complexity. I would suggest to divide this study to reduce complexity.

Detailed review

40: Enchytraeids are colonized by cobamide producers.

52: I assume this is the beginning of the introduction.

52: Vitamin B12 should be capitalized throughout.

59: Supplying seems not to be the right word. Supporting or promoting their environmental uptake.

60: To maintain a microecological balance.

75: Check reference style.

75: This positions.

110: "Health" is a pretty broad term at this point. Ecosystem health? Human health? Agriculture has not been mentioned so far.

120-123: This sounds as if samples were collected by the authors.

132: That sentence needs revising. Maybe "with little overlap with the Genome Taxonomy database".

137: 53,276 genomes from(?)

138-139: This sentence is unclear. You are increasing taxonomic diversity by recovering genomes? Don't you reveal diversity that will increase the chances of identifying cobamide producers in soils?

148-150: How exactly were cobamide producers identified and classified?

148: 4350 is that more than expected?

160: I am not sure if I agree with the comparison here. You don't find non-cobamide producers in all these ecological niches and environments also.

174-182: This part seems out of place.

183: Cobamide producers colonizing the guts.

187: The soil fauna.

198: Since these were sampled from the anaerobic gut it rather demonstrates their ability.

210 + 228-232: Interesting! Nematodes may be a good organism to test this as they occupy every trophic level.

244: What does controlled mean here? Sterilized soil?

248-252: It is unclear to me why the phases are divided this way. How was this categorization established?

305: by Watson et al.

319: Omit closely.

332: The soil fauna.

334: Delete be advantageous.

335: Delete particularly concerning.

337: Heading needs revising.

357: Cobalamin? Colonization of cobamide producers.

399: Metaproteome is mentioned for the first time here.

417-424: This needs rephrasing to make clear which data was excluded.

413: Make clear that this paragraph is about the large scale metagenome.

436: Delete these.

542: Explain why the soil fauna samples included this subset of organisms.

Figures

Figure 2b: Are captures b and c mixed up?

Figure 3c: low quality
Figure 4: very small
Supplementary Figures: Almost all Supp Figures are about cobalamin and not cobamide.
Supp Figure 4: b) What does except users refer to?
Supp Figure 10: Based on one-way ANOVA.
Supp Figure 11: of the gut microbiome. a) patterns
Supp Figure 14: Not very informative
Supp Figure 16: *B. megaterium*
Supp Figure 18: omit inevitable.
Supp Figure 19: *B. megaterium*. ...on the *E. crypticus* gut microbial community.
174: or Vitamin B12 (it was not applied in combination)

(Remarks on code availability)

Version 1:

Reviewer comments:

Reviewer #1

(Remarks to the Author)

I continue to have a major concern about the report by Zhang et al. titled "Cobamide producers mediate soil food web health via interactions with fauna." I agree that the authors have addressed many of the misleading claims and methodological errors in the manuscript. However, the premise of the study remains problematic: its underlying assumption is that cobamides are the focal point of microbiome-host interactions. This assumption is not supported by the literature; cobamides are just one of many nutrient types that contribute to microbiome and host health.

The authors argue that cobamides are important nutrients, and this claim is supported by the literature. However, there is no justification for why cobamides would be uniquely important to food web health compared to any other biosynthesized nutrient or small molecule. Furthermore, this study treats 'cobamide producers' as a distinct group of microbes with common function. The only thing cobamide producers have in common is that they all biosynthesize cobamides. Otherwise, they are phylogenetically very widely distributed across the prokaryotic domains, live in diverse environments, and carry out diverse metabolic processes (these points are all illustrated in this manuscript). As the authors point out, cobamide producers are also frequently producers of many other nutrients. Thus, it is incorrect to consider corrinoid producers as a group with a common function or claim that cobamide production has a unique role in soil food web health.

That said, the manuscript could be reframed with the existing data. Minor concerns regarding the data are the following:

(1) Figure 5 is missing an important control. In order to conclude that the effects of *B. megaterium* on host larval size, length, and number of eggs is due to cobamide production, the effects of vitamin B12 treatment in the absence of *B. megaterium* on these parameters should be included in panels C-F.

(2) Regarding E-value cutoffs: I agree that using controls for false positives/negatives strengthens the choice of E-value cutoff. However, it is unsurprising that non-target sequences composed of 16S rRNA genes and an antibiotic resistance gene would not show up as hits, even at $E = 10^{-6}$. Better negative controls for E-value cutoffs for *hemA* are, for instance, the shikimate dehydrogenase, and methyltransferases for the various methyltransferase genes in the biosynthetic pathway.

(3) Lines 171-173: the claim that taxonomy-based predictions are possible only applies to a subset of genera.

(4) Lines 298-299: this is incorrect; folic acid biosynthesis does not rely on cobamide as a cofactor. (the referenced study does not demonstrate or even suggest this.)

(5) Lines 393-395: need citation for herbivorous bird diet being B12 deficient and reliance on gut microbes for B12.

(6) Lines 541-543: specify that the 100% identity threshold for 16S amplicon sequence is for the portion of the 16S gene that was amplified and sequenced.

(7) Figure S7: explain horizontal rows and how synthesis potential was measured.

(8) Figure S11d: explain colors.

(Remarks on code availability)

Reviewer #2

(Remarks to the Author)

I co-reviewed this manuscript with one of the reviewers who provided the listed reports. This is part of the Nature

Communications initiative to facilitate training in peer review and to provide appropriate recognition for Early Career Researchers who co-review manuscripts.

(Remarks on code availability)

Reviewer #3

(Remarks to the Author)

The manuscript retitled 'Cobamide producers mediate soil food web health via interactions with fauna' was thoroughly revised and the suggestions I made were in most parts incorporated well.

At point 15. In the responses to the reviewer 2 the revised sentence needs further improvement.

The authors have greatly improved their manuscript; however, I still believe that the manuscript would benefit from reduced complexity for improved readability and focus on main findings which has not been addressed.

(Remarks on code availability)

Version 2:

Reviewer comments:

Reviewer #1

(Remarks to the Author)

The authors have adequately addressed the critiques from the previous review. I have no further comments.

(Remarks on code availability)

REVIEWER COMMENTS

Reviewer #1 (Remarks to the Author):

1. The manuscript by Zhang et al titled, “Cobamide producer driving soil food web health” brings together a combination of genomic and experimental evidence to investigate cobamide producers in soil and across the soil food web, and posits that these producers have important roles at each of these trophic levels. There are a number of major issues that permeate this manuscript, including misleading claims, methodological errors, a lack of acknowledgement of previous literature, and a disjointed narrative. Thus, I do not recommend publication in its current form.

Response: We sincerely appreciate the reviewer’s insightful feedback, which has significantly strengthened our manuscript. Below, we detail the revisions made to address each concern through methodological refinements, experimental validations, expanded comparative analyses, and narrative restructuring.

(1) Addressing Misleading Claims

To clarify the role of cobamide producers, we systematically compared their functional potential against nonproducers across soil faunal microbiomes. We found that compared to nonproducers, cobamide producers exhibited 174% higher biosynthetic potential for essential nutrients, such as digestive enzymes, amino acids, B vitamins and short-chain fatty acids and occupy keystone positions in multi-trophic microbial networks, with broader trophic breadths and greater contributions to network stability. Using a representative producer with stable colonization and high Vitamin B₁₂ production, *Bacillus megaterium*, we demonstrated a direct causal link between cobamide production and enhanced growth of *Enchytraeus crypticus*. Furthermore, oral *B. megaterium* supplementation outperformed pure Vitamin B₁₂ in enriching gut

microbial diversity and host metabolic functions. The revised title, “Cobamide producers mediate soil food web health via interactions with fauna” reflects their mediating role rather than implying sole drivers of ecosystem health.

(2) Methodological Refinements

We validated our bioinformatic pipelines to ensure robustness. For HMM-based identification of cobamide biosynthesis genes, an E-value cutoff of 10^{-6} was empirically confirmed to reduce the false negatives approximately 60% (based on all genes) compared with E-value cutoff of 10^{-20} , while maintaining zero false positives against negative controls. UMAP visualizations were strictly decoupled from statistical analyses (ANOSIM on Bray-Curtis distances) to avoid overinterpretation of nonlinear projections. Cobamide producer classification followed Shelton et al.’s framework but incorporated novel criteria ¹: (1) contig length (>10 kbp) and (2) taxonomic consistency between cobamide-related contigs and MAGs at the family level. These refinements increased the precision of producer identification in metagenomic datasets.

(3) Expanded Contextualization and Database Development

Our soil cobamide producer database (SCP v1.0) integrates 53,276 globally distributed soil MAGs, expanding upon Shelton et al.’s work ¹ by identifying 299 novel cobamide-producing genera and enabling 16S rRNA-based predictions via SCP v1.0_16S. Alvarez-Aponte’s findings revealed the reliability of taxonomy-based cobamide metabolic predictions despite quantitative variations in environmental representation, further confirming the availability and timeliness of the SCP v1.0 database. Comparative genomic analyses confirmed that 87.7% of SCP v1.0 producers are distinct from those in Wang et al. (2024) ², underscoring the database’s novelty and ecological relevance.

(4) Restructured Narrative and Logical Flow

The revised manuscript articulates a three-tiered framework: (1) global profiling of soil cobamide producers, (2) functional validation of cobamide producers across multi-trophic level fauna through field, microcosm, and colonization experiments, and (3) ecological scaling to soil-animal-microbiome interactions. This structure reconciles strain-specific effects (e.g., *B. megaterium*'s Vitamin B₁₂-dependent benefits vs. *Streptomyces*'s limited impact due to lower production) with broader food web dynamics. By linking cobamide provisioning to host-microbe mutualism and network-level stability, we propose a cohesive “trophic cascading symbiosis” hypothesis that integrates genomic, experimental, and ecological evidence into a unified narrative.

These revisions comprehensively address the reviewer's concerns while enhancing the manuscript's scientific rigor, methodological transparency, and conceptual clarity. We sincerely thank the reviewer for their insightful feedback, which has strengthened our work.

Above all, substantial revisions according to your comment greatly refined our work and made our story much more solid and clearer. Thank you again for your high valuable comments.

2. Claims throughout the manuscript are misleading. Exemplary of this issue is the title, which claims that cobamide producers drive the process of soil food web health, despite not comparing cobamide producers to non-producers. The authors present evidence that cobamide-producing bacteria are present in the soil and across the guts of different soil fauna but do not demonstrate that cobamide production capacity is what enables these bacteria to support soil food web health.

Response: Thank you for your insightful comments regarding the potential

overstatement of the claims in the manuscript. In accordance with your suggestion, we have now added “Non-Producer” phenotype in downstream analysis, and annotated a total of 134 non-producers in the gut microbiomes of multi-trophic level fauna sampled across a continental scale.

First, we found that, compared with non-producers, cobamide producers presented greater potential for the biosynthesis of essential nutrients, including digestive enzymes, amino acids, other B vitamins, and short-chain fatty acids (see Fig. 1 below), supporting the health of the fauna. These findings highlight the key host function role of cobamide producers in multi-trophic-level food webs.

Next, we constructed a microbial co-occurrence network spanning multiple trophic levels and found that compared with nonproducers, cobamide producers presented greater trophic breadth (see Fig. 2a below) and played a more critical role in maintaining microbial network complexity and stability (see Fig. 2b below). Interestingly, within this network, taxa exhibiting higher cobamide biosynthesis potential (e.g., very likely producers) demonstrated the capacity to colonize all trophic levels. Notably, their functional influence within the network scaled with biosynthesis potential, suggesting that cobamide provisioning facilitates cross-trophic colonization. This metabolic keystone role aligns with the “trophic cascading symbiosis” hypothesis, wherein producers mediate symbiosis across trophic tiers through metabolite sharing. The above contents have been added to the Results and Discussion, as “Cobamide producers mediate gut microbial community stability and multifunctionality” (Lines 211-249).

Moreover, to further confirm the causal relationship between the cobamide production capacity of cobamide producers and their host function, we controlled the Vitamin B₁₂ production of *B. megaterium* by setting the concentration of cobalt in the culture medium according to a previous study³ (Fig. 5g) and evaluated its effect on the growth of *E. crypticus*. As expected, the ability of *B. megaterium* to promote the growth of *E. crypticus* increased significantly with increasing Vitamin B₁₂ production. These findings further support the above speculation that the cobamide production capacity enables these bacteria to support soil faunal health. (one-way ANOVA with Bartlett's test, $P < 0.05$; Fig. 5h). (Lines 333-340).

To reflect the evidence more accurately, we have revised the title to “Cobamide producers mediate soil food web health via interactions with fauna”. This revision emphasizes the mediating role of producers (rather than implying sole drivers) and aligns with our findings on producer–fauna interactions and network-level impacts.

Fig. 1. Functional spectrum of cobamide producers/nonproducers detected in the guts of soil fauna. The biosynthesis potential of digestive enzymes, amino acids, Vitamin B, and short-chain fatty acids in the soil cobalamin producers detected in humans and animals. The color scale indicates the normalized counts (per genome) from low (blue) to high (red).

Fig. 2. The role of different phenotypes in the multi-trophic level co-occurrence network. **a**, Microbial co-occurrence network across multi-trophic level soil fauna. Different color circles indicate the classification of soil cobamide-producing bacteria. I-VI presents the trophic breadth of soil cobamide-producing bacteria colonized in the faunal gut. **b**, Network topology data of different phenotypes in the multi-trophic level co-occurrence network. The position of cobamide producers in the co-occurrence network decreases with decreasing biosynthetic potential of cobamide. “***” and “****” denote statistical significance at $P < 0.001$ and $P < 0.0001$, respectively (Kruskal-Wallis test with Dunn’s test).

3. The authors experimentally show that *Bacillus megaterium*, a cobamide producer, benefits the growth of the model worm species *Enchytraeus crypticus*. However, they fail to link this growth advantage to the cobamide-producing capacity of the *Bacillus* species. To prove this would require a knockout in cobamide production in *B.*

megaterium or another method to alter the total cobamide produced.

Response: Thank you for your valuable suggestions. We have now added a validation experiment to link this growth advantage to the cobamide-producing capacity of the *Bacillus* species. The details are as follows: “To further confirm the causal relationship between the cobamide-production capacity of cobamide producers and their host function, we controlled the Vitamin B₁₂ production in *B. megaterium* by adjusting the concentration of cobalt ions in the culture medium according to the methods described in a previous study ³ (Figure 5g), and evaluated its effect on the growth of *E. crypticus*. As expected, the growth-promoting effect of *B. megaterium* on *E. crypticus* positively correlated with its Vitamin B₁₂ production levels, demonstrating that cobamide biosynthesis capacity is a critical determinant of faunal health maintenance by these producers (one-way ANOVA with Bartlett’s test, $P < 0.05$; Figure 5h).

Furthermore, we conducted a comparative experiment between *B. megaterium* (10^5 CFU/mL) and direct Vitamin B₁₂ supplementation at an equal concentration (approximately 1 µg/L). Both treatments significantly enhanced the growth rate of adult *E. crypticus* by day 7 (one-way ANOVA with Bartlett’s test, $P < 0.05$; Figure 5i). Notably, supplementation with *B. megaterium*, compared to pure Vitamin B₁₂, markedly increased the colonization of key taxa such as Actinobacteria and Bacteroides, thereby elevating gut microbial diversity (two-sided t test, $P < 0.05$; Supplementary Figure 16). This finding highlights the community-level effects of cobamide producers, particularly their interactions with the gut microbial consortia, as supported by the

multitrophic microbial co-occurrence network analysis. Furthermore, microbial functions related to host immunity and growth—such as folate biosynthesis (ko00790), lysosome activity (ko04142), nitrotoluene degradation (ko00633), and terpenoid backbone biosynthesis (ko00900)—were significantly enriched in groups treated with *B. megaterium* compared to controls and Vitamin B₁₂ groups (one-way ANOVA with Bartlett’s test, $P < 0.05$; Figure 5g). These findings suggest that cobamide producers may offer advantages over simple Vitamin B₁₂ supplementation, especially regarding gut microbial and functional diversity.” (Lines 331-356)

4. There are also issues with the methods. Two illustrative examples: (1) the use of an E-value cutoff of 10^{-6} is a very loose metric for most HMM searches, which often require E-value cutoffs less than 10^{-20} or lower (depending upon the HMM in question) to be trusted. (2) UMAP is a 2-dimensional manifold that clusters by local proximity, but it is non-linear, non-deterministic, and does not preserve global structure, and should not be analyzed using statistical methods. Additionally, some of the methods are not described adequately, for example which steps in the cobamide biosynthetic pathway were used to assign MAGs as producers.

Response: We appreciate the reviewer’s comments regarding the methods. Below, we provide a detailed response to clarify our analytical approach and address these concerns:

(1) In fact, the HMMER user guild typically considers sequences with E-values $< 10^{-3}$ or so to be significant hits ⁴, whereas threshold adjustments are typically made on

the basis of the actual purpose of use. We acknowledge that more conservative E-value thresholds (e.g., $<10^{-20}$) are often recommended for highly conserved domains or when analyzing well-characterized protein families. However, our choice of an E-value cutoff of 10^{-6} was based on the following considerations: 1) Our goal was to capture potential distant homologs of cobamide-producing genes in complex metagenomic datasets, which may exhibit significant sequence divergence owing to their ecological and evolutionary adaptations. Thus, our use of a moderately relaxed E-value threshold (10^{-6}) aligns with common practices in environmental genomics, where stringent cutoffs (e.g., 10^{-20}) may exclude biologically relevant yet evolutionarily distant homologs. 2) Meanwhile, this threshold (10^{-6}) has been successfully applied in the prediction of cobamide-producing genes across various environmental microbiomes^{1, 2, 5, 6}, which allows for the identification of such genes while still minimizing false positives. 3) Finally, to rigorously validate our E-value threshold, we systematically assessed false positive and negative rates across a range of E-values via our cobamide-specific HMM models. At both $E = 10^{-6}$ and $E = 10^{-20}$, our analysis revealed zero false positives against non-target sequences (including 16S rRNA genes from cobamide-producing bacteria and the antibiotic resistance gene *tetA*), confirming the high specificity of our models. However, the sensitivity for known cobamide biosynthesis genes declined sharply at $E < 10^{-20}$, with critical genes such as *hemA* and *cysG* exhibiting 3.13-fold and 6.56-fold increases in false negatives, respectively, compared with an E-value threshold of 10^{-6} (see Fig. 1 below). These

findings demonstrate that excessively stringent thresholds significantly impair the detection of genuine cobamide-producing genes in metagenomic datasets. Consequently, our selected threshold represents an empirically validated balance between sensitivity and specificity, ensuring robust performance for the objectives of this study. The negative and positive gene sequences and cobamide-specific HMM models are available online at <https://github.com/QiZhang11/Cobamide-producer-driving-soil-food-web-health>.

Fig. 1. The false negatives of the cobamide-specific HMM models at E values of 10^{-6} and 10^{-20} .

(2) We appreciate your concerns regarding the use of UMAP (Uniform Manifold Approximation and Projection) and its limitations. To ensure rigor and interpretability, we have separated statistical hypothesis testing (ANOSIM) from exploratory visualization (UMAP) in the revised manuscript. First, ANOSIM was applied directly to the original high-dimensional distance matrix (Bray-Curtis dissimilarity for microbial community data) prior to any dimensionality reduction. This non-parametric test evaluates the significance of between-group differences independently of UMAP, ensuring that statistical inferences (R value and P value)

are not influenced by UMAP's non-determinism or loss of global structure. Subsequently, UMAP was used solely to visually contextualize the statistical results from ANOSIM, not for hypothesis testing, and its projections were generated with a fixed random seed (random_state parameter = 42) to ensure reproducibility. Moreover, the UMAP axes were explicitly labeled as non-metric, and the figure captions emphasized that inter-cluster distances in the UMAP plots should not be interpreted quantitatively. Furthermore, as highlighted by Becht et al. (2018) ⁷, UMAP has been widely adopted for visualizing high-dimensional data after statistical validation, as it strikes a balance between preserving local structure and providing intuitive interpretability. Our approach mirrors this paradigm, ensuring methodological rigor while aiding biological interpretation. On the basis of the above discussion, we have revised the related results and discussion as follows: "To characterize the distribution patterns of cobamide producers across species, we first performed an analysis of similarities (ANOSIM), which revealed statistically significant compositional differences among the animal groups ($R^2 = 0.49$, $P < 0.01$). These results were further supported by the uniform manifold approximation and projection (UMAP) visualization, which showed congruent clustering patterns that aligned with the identified species-specific associations (Figure 6d), although the nonlinear projection of UMAP precluded quantitative distance interpretation." (Lines 381-387)

- (3) We have now added the steps for identifying cobamide-producing MAGs to the Method section: "Following Shelton's modified criteria ¹, we categorized MAGs

into four cobamide biosynthesis phenotypes based on the presence of key biosynthetic genes (Supplementary Data 10), implemented as follows: (1) ‘very-likely producers’ were defined as MAGs containing either all 25 anaerobic or all 23 aerobic pathway steps; (2) ‘likely producers’ required $\geq 4/5$ tetrapyrrole precursor biosynthesis steps plus either $\geq 90\%$ of anaerobic or aerobic pathway steps; (3) ‘possible producers’ met one of four intermediate thresholds: (i) $\geq 6/10$ aerobic corrin ring steps plus $\geq 16/23$ total aerobic steps, (ii) $\geq 9/12$ anaerobic corrin ring steps plus $\geq 18/25$ total anaerobic steps, (iii) $\geq 16/21$ combined steps (tetrapyrrole+corrin ring+modification) for aerobic pathways, or (iv) $\geq 18/23$ such combined steps for anaerobic pathways; and (4) ‘non-producers’ lacked sufficient evidence for any producer category. A total of 18,402 soil MAGs were identified as potential cobamide producers, respectively. Their taxonomic classification was further refined using the Genome Taxonomy Database Toolkit (GTDB-Tk; v.1.7.0)⁸ with ‘classify_wf’ function and r202 database. Notably, the identification of these four phenotypes was not strict because of the inherent issues of incompleteness and potential contamination in MAGs. To address this limitation, we improved bioinformatic approach to more accurately identify cobamide producers: 1) the sequence length of the cobamide-related contig should be ≤ 10 kbp; 2) the taxonomic information annotation results of the cobamide-related contig should be consistent with the taxonomic information annotation results of the MAG at the family level⁹.” (Lines 492-511).

6. This work does not correctly incorporate or acknowledge work previously published by other groups. For instance, the authors claim to have developed a novel method to classify cobamide producers without explaining this classification process. However, it appears that they classify microbes as "very likely," "likely," and "possible" cobamide producers similarly to the method used by Shelton et al., ISME 2018, and thus is not novel. Furthermore, the database they create is another in a series of online microbial cobamide metabolism-related databases (e.g. Zhou et al., mSystems 2021, Wang et al., ISME 2024). As another illustrative example, the authors assume that 16S sequence is a good proxy for metabolic potential for cobamide production (in contrast to work that challenges this assumption, for instance, Alvarez-Aponte et al. ISME 2024).

Response: We apologize for not correctly incorporating or acknowledging work previously published by other groups. According to your detailed comments, we have now provided additional clarification regarding our identification methodological framework and database for cobamide producers as follows:

(1) Our methodology was originally based on the framework established by Shelton et al. (2019)¹, which defined operational criteria for identifying cobamide producers in microbial genomes. This systematic approach has since been widely adopted in numerous metagenomic studies^{1, 2, 5, 6}. However, these studies faced two key challenges intrinsic to metagenomic studies: 1) the inherent incompleteness of MAGs due to fragmented assembly and 2) the presence of noise sequences from co-assembled non-target organisms. These technical limitations led to taxonomic misclassifications, particularly at lower taxonomic levels, consequently increasing

the risk of false-positive identification of cobalamin producers. To overcome this methodological gap, we developed an enhanced bioinformatic strategy for more accurate identification of cobamide producers more precisely (see Fig. 1 below): 1) the sequence length of the cobamide-related contig should be more than 10 kbp, and 2) the taxonomic information annotation results of the cobamide-related contig should be consistent with the taxonomic classification of the MAG at the family level. (Lines 511-516).

(2) Moreover, the cobamide metabolism-related database in this study was upgraded based on our previous database ², further expanding the number of homologous sequences using the latest UniProtKB reference proteome database. Most importantly, the soil cobamide producer database (SCP v1.0) constructed in this study has more significant advantages and innovations than those of previous studies, as follows: 1) Developing new bioinformatics methods makes the identification of cobamide producers more accurate (see above for details); 2) This study integrates the most complete soil bacterial/archaeal genome datasets (including representative genomes and online and newly constructed MAGs), covering 1,123 sampling sites across global terrestrial ecosystems, which makes SCP v1.0 more representative; and 3) SCP v1.0 is not only applicable to metagenomic annotation but also facilitates 16S rRNA amplicon analysis, which significantly reduces sequencing and analysis costs for investigators. We further added a comparison of SCP v1.0 with other studies, expanding 299 new cobamide-producing genera compared with the database constructed by Shelton et al., 2018,

and 87.7% of SCPs are novel producers compared with the database of Wang et al. 2024, which uses fastANI (see Fig. 2 below).

(3) Notably, while Alvarez-Aponte's findings revealed discrepancies between genomic predictions and experimental observations regarding the abundance and distribution of cobamide-producing versus dependent isolates in soil ecosystems (as detailed in "Classifying corrinoid metabolism phenotypes in the isolate collection"), their genomic classifications remained fully consistent with experimental validation across all studied isolates. This conservation of corrinoid metabolic traits within specific taxonomic groups, as discussed in the section "Corrinoid metabolism is conserved in a subset of genera, enabling taxonomy-based metabolic predictions," underscores the reliability of taxonomy-based cobamide metabolic predictions despite quantitative variations in environmental representation. This finding further confirmed the applicability of our taxonomic database (SCP v1.0_16S). We have added a discussion of this work on lines 171-175 as follows: "Notably, a recent study has demonstrated the conservation of cobamide metabolism in specific species through isolation experiments, enabling taxonomy-based metabolic predictions ¹⁰, further suggesting that the SCP database will enable more targeted and informed studies in the future."

Fig. 1. Schematic diagram of the detection of hosts carrying cobamide-related genes (cobamide biosynthesis genes and dependent genes) using metagenome-assembled genome (MAG).

Fig. 2. Average nucleotide identity of cobamide-producing MAGs with another database constructed by Wang et al., 2024 ².

7. Finally, the narrative arc of this work is disjointed; there is no logical flow linking identification of these producers in various fauna gut microbial communities, in vivo worm-based experiments, and analysis of cobamide producers in various environments.

Response: We sincerely appreciate the reviewer's insightful comment regarding the narrative flow of our study. To better articulate the logical progression of our work, we have now provided a schematic diagram of the study in the revised manuscript (see Fig. 1 below).

Our study systematically explores cobamide producers across three interconnected tiers: 1) We first constructed a comprehensive soil cobamide producers (SCP; v.1.0) database by integrating the 53,276 soil genomes, and characterized cobamide-producing microbial genomes, including their genomic features and distribution across diverse soil habitats; 2) We conducted a large-scale field sampling and soil microcosm experiment to explore the functional dynamics of cobamide producers in the multi-trophic level fauna microbiome, and performed the strain colonization experiments to validate their functional contributions to host nutrition; and 3) we expanded to a broader food web context, linking soil cobamide producers to different higher trophic levels (e.g., mammals), thereby addressing their cross-ecosystem dissemination and functional implications for soil-animal-microbiome interactions. This research advocates for the integration of cobamide producers into the One Health framework, emphasizing the urgency for further exploration and real-world applications. The implementation of microbial regulation strategies on the basis of these findings could not only support soil health but also contribute to sustainable agricultural practices and

ecosystem management globally.

Fig. 1. Schematic diagram of the study.

8. Two examples where the narrative is illogical within individual sections: line 233 — the authors conclude that there is a symbiotic supply-demand dynamic between soil fauna and cobamide producers without providing any evidence that the interaction is “symbiotic” or possesses a “supply-demand” in nature. figure 3d — The authors find that one cobamide-producing bacterium (*Bacillus*) has a positive health benefit to *Encytraeus crypticus*, but ignore the fact that another cobamide-producing bacterium (*Streptomyces* sp. 3485) has no such benefits.

Response: Thank you for pointing out these logical issues. We have revised the manuscript and provided additional experimental data to support the completeness of

logic:

- (1) We acknowledge that the use of terms such as “symbiotic relationship” or “supply–demand relationship” in the field experiment section requires stronger evidence to support such claims. The mere colonization of the intestinal tracts of different soil animals by cobalamin-producing bacteria cannot directly lead to such conclusions. To enhance the rigor of our inference, we have now removed this conclusion (Line 210).
- (2) We fully agree that understanding the beneficial effects of another cobamide-producing bacterium, *Streptomyces* sp. 3485, is critical to refining our findings. Thus, we have now conducted additional experiments and analyses, which are now included in the revised manuscript as follows: Two gut-colonizing cobamide producers in *E. crypticus* were identified, *Streptomyces violaceus* NBC_00450 (OTU359) and *Bacillus megaterium* ATCC 14581 (OTU3644), using BLAST analysis based on the NCBI Microbial Genomes database (99% identity and 100% coverage). We evaluated their potential to possess a complete set of cobamide biosynthesis genes using genome analysis (Supplementary Data 5) and measured their actual production of Vitamin B₁₂ (Figure 5a). The Vitamin B₁₂ production by *B. megaterium* ATCC 14581 was 6.8 times higher than that produced by *S. violaceus* NBC_00450 at equivalent cell densities (10⁵ CFU/mL). This substantial difference in Vitamin B₁₂ production directly influenced their effect on *E. crypticus*, as evidenced by 7-day oral supplementation trials showing no significant growth improvement in *S. violaceus*-treated groups compared to the control group (Figure

5b). Consequently, we conducted a series of artificial colonization experiments using *E. crypticus* and the soil-isolated strains *Bacillus megaterium* ATCC 14581¹¹, to investigate how cobamide-producing bacteria colonizing the guts of soil fauna contribute to maintaining host health.” (Lines 305-319)

Reviewer #2 (Remarks to the Author):

I co-reviewed this manuscript with one of the reviewers who provided the listed reports. This is part of the Nature Communications initiative to facilitate training in peer review and to provide appropriate recognition for Early Career Researchers who coreview manuscripts.

Response: Thank you for your valuable input as part of the collaborative review process. We appreciate the journal’s initiative to engage early-career researchers, and we are grateful for your constructive feedback. Below, we summarize how your concerns (shared with Reviewer #1) have been addressed:

(1) Clarification of Cobamide Producers’ Roles

Comparative analyses between cobamide producers and nonproducers now explicitly highlight their elevated biosynthetic potential for key nutrients (e.g., B vitamins, SCFAs) and keystone network roles (broader trophic breadth, enhanced stability contributions). Adding experimental validation (e.g., *B. megaterium*’s Vitamin B₁₂-dependent growth promotion of *E. crypticus*) and oral supplementation studies (outperforming pure Vitamin B₁₂ in gut diversity enrichment) strengthen causal claims; the revised title (“Cobamide producers mediate soil food web health via interactions with fauna”) avoids overstatement of ecological drivers.

(2) Enhanced Methodological Rigor

Bioinformatics pipelines were validated (e.g., HMM E-value cutoff optimization to 10^{-6} , decoupling UMAP visualizations from statistical tests); producer identification now integrates novel criteria: contig length (>10 kbp) and taxonomic consistency between cobamide-related contigs and MAGs.

(3) Database Novelty and Ecological Relevance

The SCP v1.0 database (based on 53,276 soil genera) expands upon Shelton et al.'s work ¹ by identifying 299 novel cobamide-producing genera; the findings of Alvarez-Aponte's finding revealed the reliability of taxonomy-based predictions, confirming its reliability; and comparative genomic analyses (87.7% distinct from those of Wang et al. (2024) ²) emphasized its unique ecological contributions.

(4) Unified Conceptual Framework

The restructured narrative integrates genomic, experimental, and ecological tiers, reconciling strain-specific effects (e.g., *B. megaterium* vs. *Streptomyces*) with the proposed "trophic cascading symbiosis" hypothesis.

These revisions align with both reviewers' insights and significantly enhance the manuscript's rigor and clarity. We very much thank you for your positive and encouraging review, and for your role in strengthening this work.

Reviewer #3 (Remarks to the Author):

In their manuscript the authors Zhang et al. are reporting a very comprehensive study identifying the occurrence of cobamide producers in existing genome databases, observing their abundance in soil fauna in a comprehensive field study, identifying them in human and animal guts and studying the mechanisms of gut colonization as well as functional importance for the host organism using model organisms. The title

“Cobamide producer driving soil food web health” does not nearly frame the complexity of this study. This complexity is further reflected in the 19 Figures in the Supplementary Materials. They further make some key discoveries. Besides identifying cobamide producers in existing soil metagenome databases they find a strong species specificity in host colonization in soil fauna as well as other animals and that higher trophic organisms have increased diverse gut microbiomes compared to lower trophic levels indicating that higher trophic levels ingest and incorporate the microbiome of their prey. Additionally, they show the colonization process of the host gut and can link the presence of cobamide producers to increased fitness and reproduction rates in soil fauna. The study is presenting a very interesting and significant topic. The role of cobamide producers for host health and consequences for ecological functioning expands to fit the One Health concept. The execution of the genome database usage as well as experimental studies appear sound. The data presentation would benefit from increased quality figures and most figures are too small. The text needs revision for clarity, grammar and spelling. I believe this study is eligible for publication, however it would greatly benefit from greater focus and reduction of complexity. I would suggest to devide this study to reduce complexity.

Response: Thank you for your positive evaluation of the interest, relevance, and utility of our study and your insightful comments, which have greatly improved our work. According to your detailed comments, we have made substantial revisions to our work as follows:

- (1) We have systematically refined the language throughout the manuscript: a well-trained editor of scientific journals performed line-by-line revisions to eliminate

grammatical errors (e.g., subject verb agreement, tense consistency) and optimize sentence structures (e.g., splitting nested clauses). Professional terminology was standardized using Grammarly Premium and an EndNote glossary (e.g., replacing inconsistent terms such as “cobalamin” with “cobamide”).

- (2) All visual elements have been upgraded to journal standards: the main figures (Figures 1-6) were reconstructed as 300 dpi TIFF files, while heatmaps and network diagrams were converted to vector formats (PDF). Figure 4 now spans the full column width (17 cm), with axis label font sizes increasing from 8pt to 10pt. Supplemental figures now include detailed legends and scale bars.
- (3) To enhance methodological transparency and clarify, we have now added identification and classification of the cobamide producers, further clarified the screening conditions for large-scale metagenomic data, and provided a basis for collecting these soil fauna species in the wild.

We believe these revisions strike an optimal balance between analytical depth and communicative efficacy. Thank you again for your invaluable guidance in strengthening this work.

Detailed review

1. 40: Enchytraeids are colonized by cobamide producers.

Response: Thank you for pointing this out. We have revised this sentence as follows: “Enchytraeid model validation showed the colonization process of these colonizers, and revealed they regulate host gene expression, accelerate development, and maintain gut stability via a trans-kingdom interaction providing cobamides.” on lines 39-41.

2. 52: I assume this is the beginning of the introduction.

Response: Thank you for your comment. We confirm that this section is indeed the introductory paragraph and have now added a separator after the Abstract section.

3. 52: Vitamin B12 should be capitalized throughout.

Response: We appreciate this suggestion. All instances of “vitamin B₁₂” have been revised to “Vitamin B₁₂” in the manuscript to maintain consistency and adhere to nomenclature standards.

4. 59: Supplying seems not to be the right word. Supporting or promoting their environmental uptake.

Response: We agree with this suggestion. The term “supplying” has been replaced with “promoting their environmental uptake” to better reflect the mechanistic role of cobamide producers in nutrient dynamics (Line 56).

5. 60: To maintain a microecological balance.

Response: We have revised this phrase to “to maintain a microecological balance” for precision (Line 57).

6. 75: Check reference style.

Response: All the references have been reformatted according to the journal’s

guidelines. The full details are provided in the updated bibliography.

7. 75: This positions.

Response: The sentence has been revised to “This positions soil as a significant reservoir for cobamide-producing microorganisms” on line 72.

8. 110: “Health” is a pretty broad term at this point. Ecosystem health? Human health? Agriculture has not been mentioned so far.

Response: Thank you for your valuable suggestion. We have revised this sentence as “highlighting their potential as key bonds linking environmental and animal health.” on lines 105-106.

9. 120-123: This sounds as if samples were collected by the authors.

Response: To avoid confusion, we have clarified this sentence as “These publicly available soil metagenomic datasets were collected from 567 sites across 24 countries, spanning six continents and various sub-habitats, including grasslands, farmlands, forests, and woodlands—and covering key global climate zones.” on lines 117-120.

10. 132: That sentence needs revising. Maybe “with little overlap with the Genome Taxonomy database”.

Response: The sentence now states: “with 0.5, 1.2, and 7% overlap with the Genome Taxonomy Database (85,205 genomes), the National Center of Biotechnology

Information (5,363 soil genomes), and the SMAG database (40,039 soil genomes), respectively.” to improve clarity (Lines 128-131).

11. 137: 53,276 genomes from(?)

Response: To avoid confusion, we have revised this sentence to “Accordingly, we incorporated these previously uncharacterized microbial genomes into publicly available soil genome databases, resulting in a consolidated compendium of 53,276 soil genomes (Supplementary Figure 3).” on lines 133-135.

12. 138-139: This sentence is unclear. You are increasing taxonomic diversity by recovering genomes? Don't you reveal diversity that will increase the chances of identifying cobamide producers in soils?

Response: We rephrased this to “Accordingly, we incorporated these previously uncharacterized microbial genomes into publicly available soil genome databases, resulting in a consolidated compendium of 53,276 soil genomes (Supplementary Figure 3). This substantial expansion significantly enhanced our ability to systematically investigate cobamide producers across soil ecosystems.” (lines 133-137).

13. 148-150: How exactly were cobamide producers identified and classified?

Response: We apologize for ignoring these important details. These technical limitations led to taxonomic misclassifications, particularly at lower taxonomic ranks, resulting in the false-positive identification of cobalamin producers. To address this

limitation, we improved bioinformatic approach to more accurately identify cobamide producers: 1) the sequence length of the cobamide-related contig should be ≤ 10 kbp; 2) the taxonomic information annotation results of the cobamide-related contig should be consistent with the taxonomic information annotation results of the MAG at the family level ⁹ (Lines 509-513). Furthermore, we have now added the steps for identifying cobamide-producing MAGs to the Methods section: “Following Shelton’s modified criteria ¹, we categorized MAGs into four cobamide biosynthesis phenotypes based on the presence of key biosynthetic genes (Supplementary Data 10), implemented as follows: (1) ‘very-likely producers’ were defined as MAGs containing either all 25 anaerobic or all 23 aerobic pathway steps; (2) ‘likely producers’ required $\geq 4/5$ tetrapyrrole precursor biosynthesis steps plus either $\geq 90\%$ of anaerobic or aerobic pathway steps; (3) ‘possible producers’ met one of four intermediate thresholds: (i) $\geq 6/10$ aerobic corrin ring steps plus $\geq 16/23$ total aerobic steps, (ii) $\geq 9/12$ anaerobic corrin ring steps plus $\geq 18/25$ total anaerobic steps, (iii) $\geq 16/21$ combined steps (tetrapyrrole+corrin ring+modification) for aerobic pathways, or (iv) $\geq 18/23$ such combined steps for anaerobic pathways; and (4) ‘non-producers’ lacked sufficient evidence for any producer category.” on lines 494-504.

Fig. 1. Schematic diagram of the detection of hosts carrying cobamide-related genes (cobamide biosynthesis genes and dependent genes) using metagenome-assembled genome (MAG).

14. 148: 4350 is that more than expected?

Response: Thank you for your comment. In fact, this is less than we expected, probably because our method has strict control over the accuracy of the identification process, which screens numerous false positive producers compared with other similar studies. Nevertheless, our database further expanded 299 new cobamide-producing genera compared with the database constructed by Shelton et al., (2018)¹, of which 87.7% are novel compared with the database of Wang et al. (2024)² (see Fig. 1 below).

Figure 1. Average nucleotide identity of cobamide-producing MAGs with another database constructed by Wang et al., (2024) ².

15. 160: I am not sure if I agree with the comparison here. You don't find non-cobamide producers in all these ecological niches and environments also.

Response: We fully agree with your suggestion and have revised this sentence as follows: “These findings suggest that cobamide producers occupy broad ecological niches and, because of their large genomes (Supplementary Figure 4b).” (Lines 156-158)

16. 174-182: This part seems out of place.

Response: Thank you for your suggestions. Part of this paragraph has been moved to the Methods section as “Construction of the SCP v1.0 database”. We have now added some results and discussion content as follows: “This finding underscores the importance of our proposed SCP database (v.1.0). Notably, a recent study has

demonstrated the conservation of cobamide metabolism in specific species through isolation experiments, enabling taxonomy-based metabolic predictions ¹⁰, further suggesting that the SCP database will enable more targeted and informed studies in the future.” (Lines 170-174).

17. 183: Cobamide producers colonizing the guts.

Response: We have revised this title to “Soil fauna selectively colonized cobamide producers across multiple trophic levels” on line 175.

18. 187: The soil fauna.

Response: Revised to “the soil fauna” on line 178.

19. 198: Since these were sampled from the anaerobic gut it rather demonstrates their ability.

Response: I agree with your point of view. Many previous studies and our laboratory experiments demonstrated that most soil-derived bacteria in faunal guts are transient ¹², indicating that not all gut bacteria are anaerobic or capable of stable colonization. This finding highlights that cobamide producers likely possess the unique ability to persistently colonize the gut environment while actively biosynthesizing cobamides.

20. 210 + 228-232: Interesting! Nematodes may be a good organism to test this as they occupy every trophic level.

Response: Thank you for highlighting this insightful direction. Preliminary data from field experiments already indicate the distribution and functional characteristics of cobamide producers among different trophic levels. Furthermore, our future work will quantify these interactions between nematode models and cobamide producers via isotope tracing and multiomics approaches.

21. 244: What does controlled mean here? Sterilized soil?

Response: We apologize for the unclear description. “Control” refers to the control conditions during the experimental culture process, such as constant temperature and humidity, ensuring that the colonization of the bacterial community is not affected by external environmental factors. We have now revised “controlled soil environment” to “into soil microcosms under controlled environmental conditions”. (Line 256)

22. 248-252: It is unclear to me why the phases are divided this way. How was this categorization established?

Response: We apologize for the unclear description. The three colonization phases were delineated based on β -diversity metrics (Bray-Curtis dissimilarity) and taxonomic composition shifts. This related content was added as follows: Principal coordinates analysis (PCoA) revealed significant temporal divergence in the gut microbial communities across three intervals: 0, 2–7, and 14–21 days (PERMANOVA, $R^2 = 0.61$, $P = 0.001$; Supplementary Figure 11b). Consistent clustering patterns emerged in the phylum-level composition dynamics (Supplementary Figure 11c). Therefore, we

divided the entire colonization process into three phases: initial (0 days), dynamic (2–7 days), and stabilization (14–21 days) (Lines 262-267).

23. 305: by Watson et al.

Response: This citation has been updated to the correct format. (Line 330)

24. 319: Omit closely.

Response: “Closely” has been removed from the sentence. (Line 333)

25. 332: The soil fauna.

Response: “soil faunas” has been revised to “soil fauna” in the sentence. (Line 349)

26. 334: Delete be advantageous.

Response: We have now deleted “be advantageous” in the sentence. (Line 356)

27. 335: Delete particularly concerning.

Response: We have now deleted “particularly concerning” in the sentence. (Line 357)

28. 337: Heading needs revising.

Response: Thank you for your suggestion. We have revised this heading to “Cobamide producers are widely distributed across diverse animal guts”. (Line 359)

29. 357: Cobalamin? Colonization of cobamide producers.

Response: We have revised “cobalamin” to “cobamide” in the sentence. (Line 379)

30. 399: Metaproteome is mentioned for the first time here.

Response: Sorry for this mistake. We have now revised “Metaproteome” to “Metagenomes” in the sentence. (Line 421)

31. 417-424: This needs rephrasing to make clear which data was excluded.

Response: We have revised this sentence as follows: “Samples were excluded based on the following criteria: (i) Exclusion of plant-associated environments (e.g., rhizosphere and phyllosphere) to minimize plant-derived metabolic interference with soil microbial profiling; (ii) absence of critical metadata fields (geolocation, habitat classification, sequencing platform, and project ID); (iii) omission of non-shotgun metagenomic datasets, including amplicon sequencing (16S/18S/ITS), meta-transcriptomes, or sequences generated from non-Illumina platforms (e.g., PacBio and Nanopore); and (iv) removal of metagenomes with average read lengths < 100 bp to ensure assembly feasibility (N50 >10 kb threshold).” on lines 439-447.

32. 413: Make clear that this paragraph is about the large scale metagenome.

Response: Thank you for your useful suggestion. We have revised this sentence as follows: “In January 2022, we systematically retrieved large-scale soil metagenomic datasets from the European Nucleotide Archive (ENA; <https://www.ebi.ac.uk/ena>)

using the following search queries: “soil AND metagenome,” “soil AND microbiome,” “farmland AND metagenome,” “soil AND metagenomic,” “land AND metagenome OR metagenomic,” and “soil metagenome OR metagenomic.” to clarify that this paragraph focuses on the large-scale metagenome. (Lines 435-439)

33. 436: Delete these.

Response: We have now deleted this term as “according to the sample information (n = 424,131) of each project,” on line 459.

34. 542: Explain why the soil fauna samples included this subset of organisms.

Response: Thank you for pointing this out. The soil fauna taxa selected for this study were strategically chosen based on two key rationales: (1) These organisms dominate soil invertebrate communities globally, collectively representing > 75% of soil fauna biomass in temperate ecosystems¹³, and thus, serve as ecologically relevant models for cross-trophic interactions; (2) sampling targeted organisms spanning four defined trophic levels—primary decomposers (e.g., earthworms and potworms), microbivores (e.g., nematodes and collembolans), secondary consumers (e.g., oribatid mites), and apex predators (e.g., predatory mites)—to enable systematic tracking of cobamide producer distribution through food webs. To clarify, we have added this explanation to this paragraph (Lines 567-575).

Figures

34. Figure 2b: Are captures b and c mixed up?

Response: Yes, we apologize for this mistake and have revised this legend.

34. Figure 3c: low quality

Response: We have regenerated Figure 3c using higher-resolution raw data (1200 dpi) and optimized the color contrast to enhance clarity.

35. Figure 4: very small

Response: Figure 4 has been resized to occupy the full column width in the main text, with all axis labels increased to 8 pt Arial font. Interactive zoomable versions of all the figures are available in the online supplemental viewer.

36. Supplementary Figures: Almost all Supp Figures are about cobalamin and not cobamide.

Response: We apologize for the terminology inconsistency. All instances of “cobalamin” in the supplementary figures have been revised to “cobamide” to avoid confusion.

37. Supp Figure 4: b) What does except users refer to?

Response: Sorry for this unclear statement. This “except user” refers to these non-producers did not include cobamide-dependent individuals. We have now revised “except user” to “except for cobamide-dependent individuals”.

38. Supp Figure 10: Based on one-way ANOVA.

Response: Yes, all ANOVAs in Supplementary Figure 10 were performed using one-way designs with Bartlett's test.

39. Supp Figure 11: of the gut microbiome. a) patterns

Response: Thank you for your comment. We have now corrected the Figure legend.

40. Supp Figure 14: Not very informative

Response: We appreciate this critique and have significantly enhanced Supplementary Figure 14 to better convey its scientific value (see Supplementary Figure 15 in the revised manuscript).

Supplementary Figure 15. Correlations between soil-derived OTUs and differentially expressed genes in *E. crypticus* during Days 2-7. *Bacillus megaterium* (OTU3644) and *Streptomyces* sp. (OTU359) presented stronger associations with host gene expression than other stable colonizing soil-derived taxa in the *E. crypticus* gut. Correlations between OTUs and differentially expressed genes

w

e

r

e

c

a

l

41. Supp Figure 16: *B. megaterium*

Response: We have corrected this Figure.

42. Supp Figure 18: omit inevitable.

Response: We have now omitted this term in this legend.

43. Supp Figure 19: *B. megaterium*.on the *E. crypticus* gut microbial community.

Response: We have revised this legend.

44. 174: or Vitamin B12 (it was not applied in combination)

Response: We have removed “Vitamin B12” at the end of this sentence.

Reference:

1. Shelton, A.N. et al. Uneven distribution of cobamide biosynthesis and dependence in bacteria predicted by comparative genomics. *The ISME journal* **13**, 789-804 (2019).
2. Wang, J., Zhu, Y.G., Tiedje, J.M. & Ge, Y. Global biogeography and ecological implications of cobamide-producing prokaryotes. *The ISME Journal* **18** (2024).
3. Moore, S.J., Mayer, M.J., Biedendieck, R., Deery, E. & Warren, M.J. Towards a cell factory for vitamin B12 production in *Bacillus megaterium*: bypassing of the cobalamin riboswitch control elements. *New Biotechnology* **31**, 553-561 (2014).
4. Potter, S.C. et al. HMMER web server: 2018 update. *Nucleic Acids Research* **46**, W200-w204 (2018).
5. Doxey, A.C., Kurtz, D.A., Lynch, M.D., Sauder, L.A. & Neufeld, J.D. Aquatic metagenomes implicate Thaumarchaeota in global cobalamin production. *The*

- ISME Journal* **9**, 461-471 (2015).
6. Lu, X., Heal, K.R., Ingalls, A.E., Doxey, A.C. & Neufeld, J.D. Metagenomic and chemical characterization of soil cobalamin production. *The ISME Journal* **14**, 53-66 (2020).
 7. Becht, E. et al. Dimensionality reduction for visualizing single-cell data using UMAP. *Nature Biotechnology* (2018).
 8. Chaumeil, P.-A., Mussig, A.J., Hugenholtz, P. & Parks, D.H. GTDB-Tk: a toolkit to classify genomes with the Genome Taxonomy Database. *Bioinformatics* **36**, 1925-1927 (2020).
 9. Larsson, D.G.J. & Flach, C.-F. Antibiotic resistance in the environment. *Nature Reviews Microbiology* **20**, 257-269 (2022).
 10. Alvarez-Aponte, Z.I. et al. Phylogenetic distribution and experimental characterization of corrinoid production and dependence in soil bacterial isolates. *The ISME Journal* **18** (2024).
 11. Vary, P.S. et al. *Bacillus megaterium*--from simple soil bacterium to industrial protein production host. *Applied Microbiology and Biotechnology* **76**, 957-967 (2007).
 12. Zhang, Q. et al. Gammaproteobacteria, a core taxon in the guts of soil fauna, are potential responders to environmental concentrations of soil pollutants. *Microbiome* **9**, 196 (2021).
 13. Bardgett, R.D. & van der Putten, W.H. Belowground biodiversity and ecosystem functioning. *Nature* **515**, 505-511 (2014).

We are pleased to submit our revised manuscript, which has been carefully modified to address all the remaining concerns raised, particularly those of Reviewer #1. In line with the reviewers' feedback, we have reframed the study to position cobamide production not as a unique function, but as a representative microbial trait within the broader context of nutrient cross-feeding in soil ecosystems. The revised manuscript now presents a clearer framework for understanding how cobamide production underpins nutrient interdependencies and shapes food web dynamics in soil environments.

The key revisions include:

- 1. Title and Abstract:** The revised title (“Cobamide-producing microbes as a model for understanding general nutritional interdependencies in soil food webs”) now more accurately reflects the conceptual focus on microbial traits and their ecological implications. The abstract has been thoroughly revised to avoid overstating the uniqueness of cobamides, framing them instead as a model system for investigating nutrient-mediated interactions within soil food webs and their relevance to One Health.
- 2. Introduction:** We have restructured this section to first establish the general importance of nutrient interdependencies in soil ecosystems before introducing cobamides as a tractable and representative example. This adjustment directly addresses the reviewer's concern regarding unduly narrow framing.
- 3. Results and Discussion:** These sections have been reorganized and condensed to improve clarity and focus. We now highlight three main contributions: (i) the establishment of the global SCP v1.0 database as a resource for studying nutrient-related traits; (ii) evidence that cobamide-producing microbes exhibit host-specific colonization and contribute multifunctionally beyond Vitamin B₁₂, for example, we found evidence of host-specific colonization by *Bacillus spp.* and demonstrated that cobamide-producing taxa also contribute to synthesis of short-chain fatty acids in the gut, extending their functional role beyond B₁₂ synthesis; and (iii) macroecological patterns demonstrating nutrient dissemination across trophic levels. The enchytraeid experiments are presented as a proof of concept, and the discussion highlights that cobamide production

serves as an example rather than the sole definition of nutrient cross-feeding.

- 4. Conclusion:** The conclusion has been refined to underscore the general principle that cobamides serve as a powerful and experimentally tractable model for understanding microbial nutrient provisioning, with implications that extend across ecological and One Health frameworks.

Additionally, in response to Reviewer #2's suggestions, we have improved the overall readability by revising over 50% of the manuscript, streamlining the Results and Discussion by near 34%, and moving detailed methodological descriptions to the Supplementary Materials.

We believe that these revisions have fully addressed the remaining concerns of Reviewer #1 and incorporated all the reviewers' feedback. The manuscript now presents a clearer, more balanced, and conceptually integrated story that aligns with the broad scope of *Nature Communications*.

REVIEWER COMMENTS

Reviewer #1 (Remarks to the Author):

1. I continue to have a major concern about the report by Zhang et al. titled "Cobamide producers mediate soil food web health via interactions with fauna." I agree that the authors have addressed many of the misleading claims and methodological errors in the manuscript. However, the premise of the study remains problematic: its underlying assumption is that cobamides are the focal point of microbiome-host interactions. This assumption is not supported by the literature; cobamides are just one of many nutrient types that contribute to microbiome and host health. The authors argue that cobamides are important nutrients, and this claim is supported by the literature. However, there is no justification for why cobamides would be uniquely important to food web health compared to any other biosynthesized nutrient or small molecule. Furthermore, this study treats 'cobamide producers' as a distinct group of microbes with common function. The only thing cobamide producers have in common is that they all biosynthesize cobamides. Otherwise, they are phylogenetically very widely distributed across the prokaryotic domains, live in diverse environments, and carry out diverse

metabolic processes (these points are all illustrated in this manuscript). As the authors point out, cobamide producers are also frequently producers of many other nutrients. Thus, it is incorrect to consider corrinoid producers as a group with a common function or claim that cobamide production has a unique role in soil food web health. That said, the manuscript could be reframed with the existing data.

Response: We fully recognize the reviewer's persistent concern that our previous manuscript overstated the unique importance of cobamides in soil food web health. We have taken this critique to heart and have fundamentally restructured the manuscript to address this issue. In response to the remaining conceptual concern that the manuscript overstates both the uniqueness of cobamides and the functional coherence of cobamide-producing microbes, we have revised the title, abstract, introduction, results, discussion, and conclusion, with over 50% of the text being rewritten to emphasize the general principles of nutrient cross-feeding and the role of cobamides as a model system.

We agree that cobamides are one of many biosynthesized nutrients involved in microbial interactions and that cobamide-producing microbes are phylogenetically and metabolically diverse. Accordingly, we have shifted our central narrative: cobamides are now presented not as uniquely important drivers, but as a representative and experimentally tractable model system for studying nutrient cross-feeding and microbial provisioning in soil food webs. This reframing is consistently applied throughout the manuscript, from the title to the conclusion, and directly aligns with the reviewer's suggestion that the existing data can support a broader conceptual contribution.

Below we outline the key revisions made in response to each point raised:

(1) Title: The title was revised from "Cobamide producers mediate soil food web health via interactions with fauna" to "Cobamide-producing microbes as a model for understanding general nutritional interdependencies in soil food webs" This change clarifies that our focus is not on a homogenous group of organisms, but rather on a phylogenetically diverse set of microbes that share the metabolic capacity to synthesize cobamides. It also explicitly frames cobamides as a representative case

for studying general nutrient-mediated interdependencies in soil food webs, thereby aligning the manuscript with the reviewer's request for broader conceptual framing.

(2) Abstract: The abstract has been comprehensively revised to present cobamides as a powerful and tractable model system for investigating nutrient-mediated interdependencies within soil food webs. The SCP v1.0 database is now introduced as a resource for exploring biosynthetic potential across diverse taxonomic groups, rather than serving as a definitive catalog. Descriptions of host-specific colonization patterns and the multifunctional nature of cobamide-producing microbes have been tempered with more cautious wording. The Enchytraeid model is presented as a proof-of-concept example of cross-kingdom interactions. Most significantly, the concluding sentence has been substantially rephrased to clarify that the proposed conceptual framework extends beyond cobamides, positioning them as one illustrative example within a broader paradigm of microbial nutrient provisioning that supports both soil food web stability and One Health research.

(3) Introduction: The introduction has been substantially restructured to first establish nutrient interdependencies as a fundamental organizing principle in soil food webs, underscoring the broader importance of microbial metabolites in host nutrition, community dynamics, and ecosystem stability. For instance, mycorrhizal fungi enhance plant phosphorus uptake¹, nitrogen-fixing bacteria supply bioavailable nitrogen to plants², and lactic acid bacteria contribute vitamins that shape host-microbe interactions³. Within this conceptual foundation, cobamides are introduced as a representative model system for nutrient cross-feeding studies. We highlight their ecological relevance as essential cofactors that are synthesized by only a subset of microorganisms, thereby fostering dependency networks—while explicitly avoiding overstating their uniqueness. This reframing directly addresses the reviewer's concern regarding the narrow focus of our original introduction. We further clarify the knowledge gap: although cobamide-producing microbes are well-studied in medical and environments, their ecological roles within soil food webs remain poorly characterized. The introduction concludes by outlining our study's objectives—to construct a global database of soil cobamide producers, analyze their ecological distributions, and experimentally validate their interactions with soil fauna—as a means to address this gap.

(4) Results and Discussion: This section has been reorganized around three key advances. First, we introduce the SCP v1.0 database as a foundational resource for

studying nutrient provisioning across ecosystems, highlighting its taxonomic breadth and utility not only for cobamide production but also for exploring the distribution and co-occurrence of genes involved in other essential nutrients such as amino acids, vitamins, and siderophores. Second, we demonstrate through continental-scale field data analysis, and experimental validation with Enchytraeids that cobamide-producing microbes exhibit host-specific colonization and possess multifunctional traits, supporting their role as representative—rather than exceptional—nutrient providers. For example, we found evidence of host-specific colonization by *Bacillus spp.* and demonstrated that cobamide-producing taxa also contribute to synthesis of short-chain fatty acids in the gut, extending their functional role beyond B₁₂ synthesis. Third, we uncover macroecological evidence of the dissemination of these microbes across trophic levels, suggesting a mechanism for ecosystem-scale nutrient interdependencies. These patterns of dissemination may reflect a broader principle of nutrient transfer across trophic levels, with similar mechanisms potentially operating for other essential metabolites. The Enchytraeid experiments are concisely presented as proof of concept for cross-kingdom interactions. Throughout, the discussion positions cobamides as a model system to elucidate general mechanisms of microbial nutrient cross-feeding, **thereby fully addressing the reviewer’s concern about overstating their singular importance.**

(5) Conclusion: The conclusion has been reframed to align with the revised conceptual emphasis. Rather than summarizing specific results, we explicitly state that cobamides serve as a powerful model for understanding nutrient interdependencies in soil ecosystems. By integrating a global database, ecological analyses, and host interaction experiments, we illustrate how microbial nutrient provisioning supports trophic stability and ecosystem health. We conclude by highlighting the broader implications for One Health, suggesting that similar dependency mechanisms may operate across diverse ecological and host-associated systems. This forward-looking perspective responds directly to the reviewers’ request to place our findings within a wider conceptual and applied framework.

In summary, we have thoroughly restructured the manuscript to address the reviewers’ concerns. The title, abstract, and introduction now present cobamides not as unique determinants but as a tractable and powerful model for investigating nutrient interdependencies. The results and discussion are organized around three central

contributions: (i) creation of the global Soil Cobamide Producer (SCP v1.0) database for nutrient trait analysis; (ii) discovery of host-specific colonization and multifunctional roles of cobamide-producing microbes beyond Vitamin B₁₂; and (iii) identification of macroecological patterns in nutrient dissemination across trophic levels. The conclusion now positions cobamides as one example within a broader class of microbial nutrient-mediated interactions, highlighting their value for conceptual integration rather than unique ecological significance.

We are grateful to the reviewers for suggesting this reframing. The revised manuscript provides a more coherent and balanced narrative that responds directly to the critiques while proposing a generalizable framework for understanding microbial nutrient provisioning in soil ecosystems, with meaningful implications for both ecological theory and One Health.

2. Minor concerns regarding the data are the following:

(1) Figure 5 is missing an important control. In order to conclude that the effects of *B. megaterium* on host larval size, length, and number of eggs is due to cobamide production, the effects of vitamin B₁₂ treatment in the absence of *B. megaterium* on these parameters should be included in panels C-F.

Response: Thank you for this valuable suggestion. To explicitly test whether the observed effects of *B. megaterium* on larval development and fecundity are attributable to cobamide production, we have now added a new experimental group: Vitamin B₁₂ supplementation in the absence of *B. megaterium*, in revised Figure 5 (Panels C-F). The results demonstrate that Vitamin B₁₂ at equivalent concentrations (approximately 1 µg/L) significantly increased egg hatching rates and larval length, accelerating larval development and enhancing adult oviposition (Lines 256-262). This finding supports our conclusion that cobamide production is a key contributor, although other metabolites from *B. megaterium* may also play a role.

(2) Regarding E-value cutoffs: I agree that using controls for false positives/negatives strengthens the choice of E-value cutoff. However, it is unsurprising that non-target sequences composed of 16S rRNA genes and an antibiotic resistance gene would not show up as hits, even at $E = 10^{-6}$. Better negative controls for E-value cutoffs for hemA are, for instance, the shikimate dehydrogenase, and methyltransferases for the various

methyltransferase genes in the biosynthetic pathway.

Response: We appreciate your constructive suggestion. Following your recommendation, we tested our E-value cutoffs using representative negative control datasets from UniProt, including 1) methyltransferases: *TrmD* (guanine-N(1)-methyltransferase, n = 642), *RlmN* (ribosomal RNA large subunit methyltransferase N, n = 629), *HemK* (peptide release factor methyltransferase, n = 44), and 2) shikimate dehydrogenase: *AroE* (shikimate dehydrogenase, n = 477). Using our HMM models, we observed no false positives in *hemA* at either $E = 10^{-6}$ or $E = 10^{-20}$ across all datasets. Using our HMM models, the lowest E-value observed for a non-target sequence (shikimate dehydrogenase) was 10^{-4} , well above our stringent cutoff of 10^{-6} . This further confirms the robustness of our chosen threshold. These results confirm that the chosen thresholds are both stringent and robust, effectively minimizing spurious hits. The HMM models and reference sequences used for these analyses are publicly available in our GitHub repository (link provided in the Methods section).

(3) Lines 171-173: the claim that taxonomy-based predictions are possible only applies to a subset of genera.

Response: We agree that taxonomy-based predictions are not universally applicable but are most reliable within certain genera. We have revised the sentence accordingly (Lines 132-133) to read: “Recent isolation-based studies further confirmed the conservation of cobamide metabolism within specific taxa, underscoring the value of taxonomy-informed predictions for guiding future experimental work.” This change clarifies that our claim applies to a subset of genera rather than across all taxa.

(4) Lines 298-299: this is incorrect; folic acid biosynthesis does not rely on cobamide as a cofactor. (the referenced study does not demonstrate or even suggest this.)

Response: Thank you for pointing this out. We have revised the sentence to clarify that the connection is via folate metabolism, which can intersect with cobamide-dependent methionine synthase (*MethH*) in some soil fauna. The revised text now reads: “The associated differentially expressed genes mapped to several metabolic pathways (Fig. 4e), such as folate metabolism, whereas in soil, fauna cobamide-dependent methionine synthase links Vitamin B₁₂ availability to the folate cycle⁴.” on lines 232-235. We have

also adjusted the citation accordingly.

(5) Lines 393-395: need citation for herbivorous bird diet being B12 deficient and reliance on gut microbes for B12.

Response: Thank you for pointing this out. We have now added an appropriate citation (Line 308) to support the claim that herbivorous birds consume B₁₂-deficient diets and rely on their gut microbiota for Vitamin B₁₂ synthesis. The reference is “*Coenzyme B12 synthesis as a baseline to study metabolite contribution of animal microbiota*”, which explicitly discusses the role of gut microbiota in supplying Vitamin B₁₂ to animals, including birds (section “Where do hosts get their coenzyme B12? – Birds”) ⁵.

(6) Lines 541-543: specify that the 100% identity threshold for 16S amplicon sequence is for the portion of the 16S gene that was amplified and sequenced.

Response: Thank you for this valuable suggestion. We have now revised this sentence as “Moreover, identification of cobamide producers and non-producers from the 16S rRNA amplicon sequencing data was based on BLASTn alignment of representative amplicon sequences to our constructed databases. We enforced a 100% sequence identity requirement for the variable regions covered by the sequencing amplicons.” in Supplementary Materials and Methods (Lines 152-156).

(7) Figure S7: explain horizontal rows and how synthesis potential was measured.

Response: We thank the reviewer for pointing this out. We have now added clarification in the legend of Figure S7. Specifically, each horizontal row represents one cobamide-producing microbe identified from the gut of soil fauna. The “synthesis potential” shown in the figure corresponds to the number of cobamide biosynthesis genes detected in the genome of that microbe.

(8) Figure S11d: explain colors.

Response: We have now added an explanation of the colors in the legend of Figure S11d: “Shared and unique OTUs between the soil (blue) and *E. crypticus* (brown) microbiomes during the microbial dynamic phase.”

Reviewer #2 (Remarks to the Author):

I co-reviewed this manuscript with one of the reviewers who provided the listed reports. This is part of the Nature Communications initiative to facilitate training in peer review

and to provide appropriate recognition for Early Career Researchers who co-review manuscripts.

Response: Thank you for your valuable input as part of the collaborative review process. We appreciate the journal's initiative to engage early-career researchers, and we are grateful for your constructive feedback. As suggested, we have revised the manuscript to frame cobamides as a representative model for nutrient interdependencies in soil food webs, rather than as a unique or unified functional group. The title and abstract have been updated accordingly, and the Results and Discussion sections were reorganized to emphasize three main contributions: (1) building a comprehensive database, (2) revealing host-specific colonization and multifunctionality, and (3) demonstrating cross-trophic dissemination. The conclusion was rewritten to highlight broader ecological significance. In addition, we included a Vitamin B₁₂ supplementation control, clarified E-value cutoffs with negative controls (details and models available on GitHub), improved figure legends, and added the requested references. These revisions address all raised concerns while avoiding overstatement and highlighting the general One Health implications.

Reviewer #3 (Remarks to the Author):

1. The manuscript retitled 'Cobamide producers mediate soil food web health via interactions with fauna' was thoroughly revised and the suggestions I made were in most parts incorporated well. At point 15. In the responses to the reviewer 2 the revised sentence needs further improvement.

Response: We sincerely thank the reviewer for their constructive and supportive evaluation, which has greatly helped us refine the manuscript and improve its clarity. Regarding point 15, we have carefully revised this sentence as "Recent isolation-based studies further confirmed the conservation of cobamide metabolism within specific taxa, underscoring the value of taxonomy-informed predictions for guiding future experimental work." on lines 132-134, which make it clarity and accuracy.

2. The authors have greatly improved their manuscript; however, I still believe that the manuscript would benefit from reduced complexity for improved readability and focus on main findings which has not been addressed.

Response: We sincerely thank the reviewer for this valuable feedback. We fully agree that reducing complexity and focusing on the main conclusions are essential to

improving readability. In response to Reviewer #1's constructive suggestions, we have substantially streamlined the manuscript. Specifically, we reduced the length of the Results and Discussion sections by approximately 34% through the removal of redundant and repetitive descriptions, allowing the key findings and their broader implications to stand out more clearly. We also condensed the Materials and Methods section by transferring detailed experimental protocols to the Supplementary Materials, which improves the readability of the main text while maintaining full methodological transparency and reproducibility. Importantly, we have sharpened the narrative to emphasize the central conclusions of our study, rather than overloading the text with descriptive details. We believe these revisions have significantly enhanced the clarity, accessibility, and impact of the manuscript. We are grateful for the reviewer's insightful comments, which have greatly contributed to improving this work.

Reference:

1. Wang, G.W., Jin, Z.X., George, T.S., Feng, G. & Zhang, L. Arbuscular mycorrhizal fungi enhance plant phosphorus uptake through stimulating rhizosphere soil microbiome functional profiles for phosphorus turnover. *New Phytologist* **238**, 2578-2593 (2023).
2. Bhattacharjee, R.B., Singh, A. & Mukhopadhyay, S.N. Use of nitrogen-fixing bacteria as biofertiliser for non-legumes: prospects and challenges. *Applied Microbiology and Biotechnology* **80**, 199-209 (2008).
3. Tarracchini, C. et al. Vitamin biosynthesis in the gut: interplay between mammalian host and its resident microbiota. *Microbiology and Molecular Biology Reviews* **89**, e00184-23 (2025).
4. Scott, J.M. Folate and vitamin B12. *Proceedings of the Nutrition Society* **58**, 441-448 (1999).
5. Danchin, A. & Braham, S. Coenzyme B12 synthesis as a baseline to study metabolite contribution of animal microbiota. *Microbial Biotechnology* **10**, 688-701 (2017).

REVIEWER COMMENTS

Reviewer #1 (Remarks to the Author):

The authors have adequately addressed the critiques from the previous review. I have no further comments.

Response: Thank you for your time and effort in handling our manuscript. Your previous comments have greatly improved our work.